# DISK: DOMAIN INFERENCE FOR DISCOVERING SPURIOUS CORRELATION WITH KL-DIVERGENCE

## ABSTRACT

Existing methods utilize domain information to address the subpopulation shift issue and enhance model generalization. However, the availability of domain information is not always guaranteed. In response to this challenge, we introduce a novel end-to-end method called DISK. DISK discovers the spurious correlations present in the training and validation sets through KL-divergence and assigns spurious labels (which are also the domain labels) to classify instances based on spurious features. By combining spurious labels $y_s$ with true labels $y$, DISK effectively partitions the data into different groups with unique data distributions $\mathbb{P}(\mathbf{x}|y, y_s)$. The group partition inferred by DISK then can be seamlessly leveraged to design algorithms to further mitigate the subpopulation shift and improve generalization on test data. Unlike existing domain inference methods, such as ZIN (Lin et al., 2022) and DISC (Wu et al., 2023), DISK reliably infers domains without requiring additional information. We extensively evaluated DISK on different datasets, considering scenarios where validation labels are either available or unavailable, demonstrating its effectiveness in domain inference and mitigating subpopulation shift. Furthermore, our results also suggest that for some complex data, the neural network-based DISK may have the potential to perform more reasonable domain inferences, which highlights the potential effective integration of DISK and human decisions when the (human-defined) domain information is available. Codes of DISK are available at https://anonymous.4open.science/r/DISK-E23A/.

## 1 INTRODUCTION

Subpopulation shift is a common phenomenon in various real-world machine learning applications where both training and test share the same subpopulations but differ in subpopulation probabilities (Barocas & Selbst, 2016; Bickel et al., 2007). This phenomenon poses significant challenges for Empirical Risk Minimization (ERM) in practical scenarios. When ERM is applied solely based on the training dataset, it frequently encounters difficulties in generalizing to test sets exhibiting subpopulation shifts, resulting in substantial performance degradation (Shi et al., 2021; Han et al., 2022). For example, the CMNIST dataset in Figure 1 has two domains (red and green) and two classes (0 and 1). In training, the class 0 ratio is 8:2 (red:green) and for class 1, it's 2:8. In testing, the ratios shift to 1:9 for class 0 and 9:1 for class 1. This subpopulation shift causes models to learn spurious correlations, like red-0 and green-1, which don't apply in the testing set.

Numerous methods have been proposed to encourage models to learn invariant features in order to mitigate the subpopulation shift issue (Sagawa et al., 2019; Xu et al., 2020; Kirichenko et al., 2022; Shi et al., 2021; Liu et al., 2021a). These methods rely on the availability of domain information, which is commonly assumed to correlate with spurious features (Yao et al., 2022). However, practical acquisition can be challenging due to limited prior knowledge about spurious features (Creager et al., 2021; Liu et al., 2021b; Lin et al., 2022). For example, whether the color or the digit shape of the CMNIST data corresponds to the spurious feature cannot be determined.

Existing methods for inferring domain information have notable limitations. For instance, methods like EIIL (Creager et al., 2021) and LfF (Nam et al., 2020) struggle to reliably infer domain information in heterogeneous data without prior invariant information. Consider two datasets CMNIST (COLOR-MNIST) and MCOLOR (MNIST-COLOR), both containing identical data;

however, in CMNIST, color signifies domain information, while digits shape remains invariant, whereas in MCOLOR, the roles are reversed, with shape as the domain and color as the invariant. EIIL and LfF rely on either color or shape as the invariant feature to infer the domain. However, for datasets like CMNIST and MCOLOR, where data is the same, and invariant information is unknown, EIIL and LfF would fail on at least one of them (Lin et al., 2022). Approaches like DISC (Wu et al., 2023) and ZIN (Lin et al., 2022) require extra annotations or the construction of the concept bank with potential spurious features for domain inference, posing practical challenges. For example, ZIN ignores color in its standard annotations, limiting domain inference in CMNIST. Both ZIN and DISC require specific data information, which makes them less suitable as general frameworks.

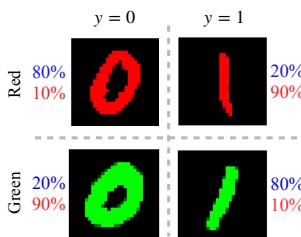

Figure 1: CMNIST with two domains. Digit color is used as domain information which is spuriously correlated with training labels. The varying probabilities in the four groups between training and testing datasets imply the existence of subpopulation shift.

Even when general data information exists, collecting additional data details reduces their efficiency compared to purely data-driven domain inference methods.

In this paper, we introduce a novel method called **D**omain **I**nference for discovering **S**purious Correlation with **K**L-Divergence (DISK). It aims to maximize the difference between the distributions of the training and validation datasets to detect spurious features and infer domain information that is highly correlated with these spurious features. DISK assigns spurious labels (also domain labels) to instances, combines them with true labels for group partitioning, and uses group-based enhancement techniques to improve generalization on the test set. As an end-to-end approach, DISK seamlessly integrates its inferred domain information with downstream methods to mitigate subpopulation shift. Importantly, DISK only requires an additional validation data for stable domain inference, eliminating the need for collecting extra information. We thoroughly explore scenarios in which validation labels are either available or unavailable and demonstrate the effectiveness of DISK in domain inference and the alleviation of the subpopulation shift issue through extensive experiments. Our contributions can be summarized as follows:

1. We propose DISK, a novel and effective end-to-end method for domain inference, that can be effectively employed to mitigate subpopulation shift and improve generalization in the test domain in Section 3. In particular, we design a KL-divergence-based objective for training the DISK domain classifier, which maximizes the difference between "spurious correlations" of the domain predictions for training data and (unlabeled) validation data. Notably, DISK only requires (unlabeled) validation data to perform domain inference, without any additional information, thus can be performed in a purely data-driven manner.
2. We introduce a simple yet effective metric for assessing the performance of domain partitioning and demonstrate the effectiveness of DISK on multiple datasets in Section 4. Besides, when further integrating DISK with the simple subsampling and retraining approach, we can achieve nearly matching or even slightly better test performance compared with the methods that explicitly rely on the true domain information. This justifies the effectiveness of DISK in mitigating the subpopulation shift when the domain information is absent.
3. We provide new insights on domain inference, illustrating that when spurious features contain complex information, the neural network-based DISK has greater potential to capture the essence of the data than human decisions (section 4.2.2). DISK partitions domains more based on the underlying similarities in patterns. This finding underscores the potential for effectively integrating DISK with human decision-making to achieve accurate domain inference in complex settings.

## 2 RELATED WORK

Many domain generalization methods utilize domain information to mitigate the issue of data distribution shift. These methods include invariant learning, which aims to boost the correlation between invariant representations and labels, thereby generating predictors that remain unaffected by different domains (Peters et al., 2016; Koyama & Yamaguchi, 2020). For instance, IRM (Arjovsky et al., 2019) and its variant IB-IRM (Ahuja et al., 2021) try to identify predictors that perform consistently well across all domains through regularization. LISA (Yao et al., 2022) acquires domain-invariant

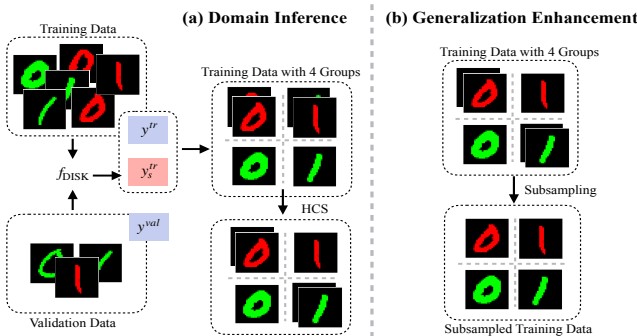

Figure 2: (a) DISK discovers the spurious correlation between the training and validation data, assigning spurious labels $y_s^{tr}$ to training instances. Subsequently, the training set is partitioned into different groups with distinct distributions $\mathbb{P}(\mathbf{x}_s, \mathbf{x}_v | g)$ where $g = (y, y_s)$. The HCS operation aids DISK in achieving a more precise inference of minority groups; (b) The training data from different domains undergo enhancement techniques, such as subsampling, to achieve a balanced training dataset with equal-sized groups for further training.

predictors by selectively combining samples with matching labels but differing domains or matching domains but differing labels, using data interpolation. Additionally, a series of Group Robustness methods are designed to address generalization challenges arising from subpopulation shift. For example, group DRO (Sagawa et al., 2019) directly optimizes performance in the worst-case group scenario through distributionally robust optimization. Some works Nam et al. (2022); Sohoni et al. (2021) proposed semi-supervised methods aimed at improving the test performance in scenarios where group labels are provided for a small fraction of the training data. Various other methods, including reweighting (Sagawa et al., 2020), regularization (Cao et al., 2019), and downsampling (Kirichenko et al., 2022), are employed to achieve a balance in the representation of both majority and minority groups. Notably, the simple yet effective downsampling method, DFR (Kirichenko et al., 2022), utilizes domain information to downsample and obtain a small, balanced dataset for retraining the final layer of the classification model.

When domain information is unavailable, EIIL (Creager et al., 2021) incorporates domain inference to directly identify domains that provide the most valuable information for downstream invariant learning. However, it relies solely on the training dataset and requires the invariant information, leading to instability in detecting spurious features. ZIN (Lin et al., 2022), when supplemented with auxiliary information like timestamps for time-series data, meta-annotations for images, and geographic data such as latitude and longitude, improves domain information inference. Acquiring such auxiliary information poses similar challenges to acquiring domain information, and the lack of prior knowledge limits ZIN's universal adaptability. Similarly, DISC (Wu et al., 2023) assists in inferring domain information by constructing a concept bank with potential spurious features, yet it encounters similar practical challenges as ZIN.

## 3 METHOD

In this section, we begin by outlining the problem setup and important notations in Section 3.1. Following that, we provide a formal definition for spurious labels in Section 3.2. Next, we discuss our method, DISK, including the optimization functions in scenarios with and without validation labels in Section 3.3. Lastly, Section 3.4 elucidates the seamless integration of DISK's inferred domain information into design algorithms to improve model generalization on test data.

### 3.1 PRELIMINARIES

Consider the dataset $\mathcal{D}$, which comprises $n$ data point-label pairs, denoted as $\mathcal{D} = \{(\mathbf{x}_i, y_i)\}_{i=1}^n$. The data $\mathbf{x}$ can be decomposed into invariant features $\mathbf{x}_v$ and spurious features $\mathbf{x}_s$. Invariant features $\mathbf{x}_v$ capture the genuine causal relationship between the data $\mathbf{x}$ and the label $y$, whereas spurious features $\mathbf{x}_s$ are typically correlated with the class label but often lack generalizability. To represent the features extracted from $\mathbf{x}$, we employ a feature extractor denoted as $\Phi$, yielding $\mathbf{z} = \Phi(\mathbf{x})$. It is expected that the representation $\mathbf{z}$ contains valuable information relevant to $y$.

We adopt the unified definition of subpopulation shift proposed by Yang et al. (2023) and consider group-based spurious correlations as defined by Sagawa et al. (2019), where the subpopulations (also groups) are defined based on the attribute (domains) and labels. The training distribution is a mixture of $K$ subpopulations, represented as $\mathbb{P}^{tr} = \sum_k^K r_k^{tr} \mathbb{P}_k(\mathbf{x}_v, \mathbf{x}_s)$, where $r_k^{tr}$ defines the mixture probabilities within the training set, and the training subpopulation is defined as $D^{tr} = \{k : r_k^{tr} > 0\}$. Similarly, the test distribution is also a mixture of $K$ subpopulations, given by $\mathbb{P}^{ts} = \sum_k^K r_k^{ts} \mathbb{P}_k(\mathbf{x}_v, \mathbf{x}_s)$, and the test subpopulation is correspondingly defined as $D^{ts} = \{k : r_k^{ts} > 0\}$. In subpopulation shift, the test set includes subpopulations observed in the training set, although with varying proportions of each subpopulation, denoted as $D^{ts} \subseteq D^{tr}$, but with $\{r_k^{ts}\} \neq \{r_k^{tr}\}$. Without domain information, it's impossible to partition the data into different groups, making it challenging to enhance generalization.

## 3.2 FORMAL DEFINITION OF SPURIOUS LABELS

In this section, we formally introduce the concept of spurious labels. Given the data $\mathbf{x}$, the label $y$, invariant features $\mathbf{x}_v$, and spurious features $\mathbf{x}_s$, alongside the data representation $\mathbf{z}$, which includes both the spurious representation $\mathbf{z}_s$ and the invariant representation $\mathbf{z}_v$, we give the following definition:

**Definition 1.** *(**Spurious Labels**) The spurious label, denoted as $y_s$, is determined by assigning labels to instances only based on the spurious representation $\mathbf{z}_s$.*

For example, in CMNIST, $\mathbf{z}_s$ represents the color (spurious feature) representation, $y_s$ represents the spurious labels assigned to instances based solely on the color representation. Since domain information is typically assumed to be spuriously correlated with the true label (Yao et al., 2022), the spurious representation-based label $y_s$, can be considered as the domain label. Therefore, each group $g$ is jointly determined by both spurious labels and true labels, i.e., $g = (y, y_s)$. In the case of CMNIST, color (red or green) serves as both the domain information and the spurious feature, with corresponding labels representing the spurious labels (also the domain labels), denoted as $y_s = \{red, green\}$. When combined with the true labels $y$ and $y_s$, CMNIST is categorized into four groups: $\{g_1, g_2, g_3, g_4\} = \{(0, red), (0, green), (1, red), (1, green)\}$ as shown in Figure 2. Dividing these groups allows the application of group-based domain generalization techniques to address subpopulation shift.

## 3.3 DOMAIN INFERENCE BASED ON SPURIOUS CORRELATION WITH KL-DIVERGENCE

To obtain the spurious label $y_s$, we introduce a novel method: **D**omain **I**nference based on **S**purious Correlation with **K**L-Divergence (DISK).

Consider three datasets that conform to subpopulation shift: the training set $\mathcal{D}^{tr}$, the validation set $\mathcal{D}^{val}$, and the test set $\mathcal{D}^{ts}$. Spurious correlation (Jackson & Somers, 1991; Haig, 2003; Yao et al., 2022; Deng et al., 2023) results in a strong association between the spurious label $y_s$ and the true label $y$ in $\mathcal{D}^{tr}$, whereas this correlation is weak or even absent in $\mathcal{D}^{val}$. By using KL-divergence $\mathrm{KL}(\cdot||\cdot)$ and mutual information $I(\cdot, \cdot)$, DISK aims to find the spurious label by (1) maximizing the correlation between the true label $y$ and spurious label $y_s$ in training set $\mathcal{D}^{tr}$; and (2) minimizing such correlation in validation set $\mathcal{D}^{val}$. In particular, the first objective can be conducted by maximizing the mutual information between $y$ and $y_s$ (denoted as **Correlation Term**), and the second objective will be performed by maximizing the discrepancy between the spurious correlations in the training set $\mathcal{D}^{tr}$ and the validation set $\mathcal{D}^{val}$ (denoted as **Spurious Term**). We employ a spurious classifier $f_{\mathrm{DISK}}$, which is designed to classify instances based on spurious representation to estimate the spurious label $y_s$, and the detailed design of our training objective is provided as follows:

**Correlation Term.** In order to encourage the correlation between the true label and spurious label in the training set, we consider the following optimization objective:

$$\max_{\mathbf{w}} I(y^{tr}; \hat{y}_{s,\mathbf{w}}^{tr}), \tag{1}$$

where the estimated spurious label $\hat{y}_{s,\mathbf{w}}^{tr} = f_{\mathrm{DISK}}(\mathbf{z}^{tr}; \mathbf{w})$ and the $\mathbf{w}$ denotes the model parameter of the spurious classifier $f_{\mathrm{DISK}}$. The representation $\mathbf{z}^{tr}$ refers to the last-layer output of the pretrained model (the model trained on the original training dataset).

**Spurious Term.** In order to maximize the discrepancy between the correlations (e.g., the correlation between $y$ and $y_s$) in the training and validation set, we consider applying the KL divergence between their corresponding conditional distributions $\mathbb{P}(y|\hat{y}_s)$, leading to the following objective for predicting the spurious label:

$$\max_{\mathbf{w}} \mathrm{KL}(\mathbb{P}(y^{tr}|\hat{y}_{s,\mathbf{w}}^{tr})||\mathbb{P}(y^{val}|\hat{y}_{s,\mathbf{w}}^{val})), \tag{2}$$

where the estimated spurious labels $\hat{y}_{s,\mathbf{w}}^{tr} = f_{\mathrm{DISK}}(\mathbf{z}^{tr}; \mathbf{w})$ and $\hat{y}_{s,\mathbf{w}}^{val} = f_{\mathrm{DISK}}(\mathbf{z}^{val}; \mathbf{w})$. Like $\mathbf{z}^{tr}$, $\mathbf{z}^{val}$ corresponds to the last linear layer's output of the pretrained model when given validation data.

**Overall Objective.** By combining (1) and (2), we derive the overall objective of DISK as follows:

$$\max_{\mathbf{w}} I(y^{tr}; \hat{y}_{s,\mathbf{w}}^{tr}) + \gamma \mathrm{KL}(\mathbb{P}(y^{tr}|\hat{y}_{s,\mathbf{w}}^{tr})||\mathbb{P}(y^{val}|\hat{y}_{s,\mathbf{w}}^{val})), \tag{3}$$

where $\gamma > 0$ is a weighting parameter used to balance the Correlation Term and the Spurious Term.

However, the overall objective of DISK in (3) faces certain issues in practical implementation. Firstly, the mutual information term is difficult to accurately estimate (Paninski, 2003; Belghazi et al., 2018). Secondly, the availability of the true label $y_{val}$ for the validation set is not always guaranteed, thus the KL divergence term cannot be calculated tractably.

To accurately compute the mutual information $I(y^{tr}; \hat{y}_{s,\mathbf{w}}^{tr})$, we demonstrate in Appendix 1 that maximizing this mutual information can be transformed into minimizing the cross-entropy $H(y^{tr}, \hat{y}_{s,\mathbf{w}}^{tr})$. This conclusion aligns with intuition because maximizing mutual information between $y^{tr}$ and $\hat{y}_{s,\mathbf{w}}^{tr}$ essentially encourages a closer alignment of their distributions, which is consistent with the objective of minimizing cross-entropy $H(y^{tr}, \hat{y}_{s,\mathbf{w}}^{tr})$. Therefore, when we have access to the validation set labels $y^{val}$, we can reformulate the overall objective of DISK as follows:

$$\min_{\mathbf{w}} H(y^{tr}, \hat{y}_{s,\mathbf{w}}^{tr}) - \gamma \mathrm{KL}(\mathbb{P}(y^{tr}|\hat{y}_{s,\mathbf{w}}^{tr})||\mathbb{P}(y^{val}|\hat{y}_{s,\mathbf{w}}^{val})). \tag{4}$$

In more typical scenarios, when the label $y^{val}$ of the validation set are unavailable or when the validation set is sampled from an unlabeled test set, computing $\mathrm{KL}(\cdot||\cdot)$ in (4) becomes impractical. To address this, we replace $y^{val}$ in $\mathrm{KL}(\cdot||\cdot)$ with the representation $\mathbf{z}^{val}$, which strongly correlates with $y^{val}$ and is always accessible. We present the following theorem:

**Theorem 1.** *[Lower Bound of Spurious Term without Accessible $y^{val}$] Given representations $\mathbf{z}^{tr}$ and $\mathbf{z}^{val}$, the spurious term is lower bounded by the following expression as:*

$$\mathrm{KL}(\mathbb{P}(y^{tr}|\hat{y}_{s,\mathbf{w}}^{tr})||\mathbb{P}(y^{val}|\hat{y}_{s,\mathbf{w}}^{val})) \geq \mathrm{KL}(\mathbb{P}(\mathbf{z}^{tr}|\hat{y}_{s,\mathbf{w}}^{tr})||\mathbb{P}(\mathbf{z}^{val}|\hat{y}_{s,\mathbf{w}}^{val})) \tag{5}$$

As stated in Theorem 1, when the label of the validation data $y^{val}$ is missing, we resort to maximizing $\mathrm{KL}(\mathbb{P}(\mathbf{z}^{tr}|\hat{y}_{s,\mathbf{w}}^{tr})||\mathbb{P}(\mathbf{z}^{val}|\hat{y}_{s,\mathbf{w}}^{val}))$ as an alternative for maximizing $\mathrm{KL}(\mathbb{P}(y^{tr}|\hat{y}_{s,\mathbf{w}}^{tr})||\mathbb{P}(y^{val}|\hat{y}_{s,\mathbf{w}}^{val}))$. We point out that maximizing a lower bound is meaningful as it provides the worst-case guarantee over the original objective. The detailed proof of Theorem 1 is provided in Appendix A.2. Therefore, when validation labels $y^{val}$ are unavailable, the overall objective of DISK can be redefined as follows:

$$\min_{\mathbf{w}} H(y^{tr}, \hat{y}_{s,\mathbf{w}}^{tr}) - \gamma \mathrm{KL}(\mathbb{P}(\mathbf{z}^{tr}|\hat{y}_{s,\mathbf{w}}^{tr})||\mathbb{P}(\mathbf{z}^{val}|\hat{y}_{s,\mathbf{w}}^{val})). \tag{6}$$

We employ the MINE algorithm (Belghazi et al., 2018) to estimate the $\mathrm{KL}(\cdot||\cdot)$ terms in (4) and (6).

### 3.4 MITIGATING SUBPOPULATION SHIFT WITH DISK

In this section, we show how to leverage DISK to mitigate subpopulation shift. As shown in Figure 2, the inferred spurious labels $y_s^{tr}$ from DISK and the true labels $y$ divide the data space into multiple groups, each characterized by a distinct distribution $\mathbb{P}(\mathbf{x}_s, \mathbf{x}_v|y, y_s)$. Based on the predicted group information from DISK, we are able to apply the existing domain generalization methods, which require the domain information of the data, to improve the generalization in the test domain. In this work, we primarily employ the Subsampling strategy (Kirichenko et al., 2022; Wu et al., 2023), which downsamples the original training dataset according to their group information (predicted

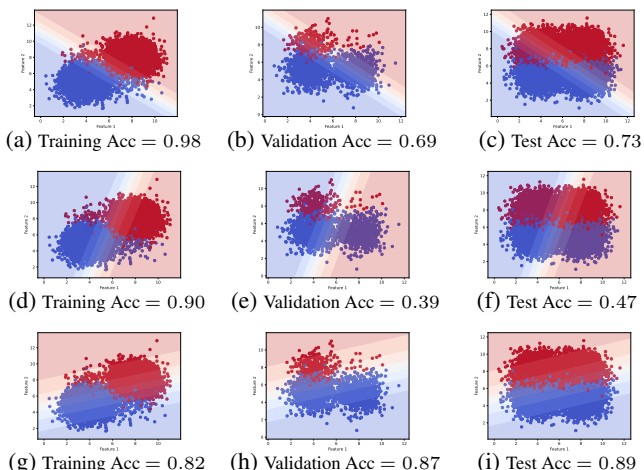

Figure 3: Illustration of the decision boundaries obtained by $f_{\text{vanilla}}$, $f_{\text{DISK}}$, $f_{\text{DISK}}$. True decision boundaries for spurious features and invariant features are vertical and horizontal respectively. (a-c) Decision Boundary and Prediction Accuracy for $f_{\text{vanilla}}$. (d-f) DISK with Accessible $y^{val}$: Decision Boundary and Prediction Accuracy for $f_{\text{DISK}}$. (g-i) DISK with Subsampling: Decision Boundary and Prediction Accuracy for $f_{\text{DISK}}$.

by DISK), such that all groups are balanced. Then, we can proceed to retrain a model using the subsampled dataset [1].

Intuitively, in the subsampled dataset, spurious features are distributed evenly within each group, so that their correlations with the labels in the training dataset can be eliminated. However, since the domain inference performance of DISK may not be perfect, directly applying the raw predictions of DISK cannot guarantee balanced spurious features in the subsampled dataset. The reason behind the imperfect domain inference is that the (true) groups in the original dataset are extremely unbalanced. For instance, in CMNIST dataset, the sample size of red digits with label 1 (or green digits with label 0) is much smaller than that of red digits with label 0 and green digits with label 1 (see Figure 1). Such minority groups may be difficult to be perfectly identified by DISK compared to the majority groups, which further affects the balance (with respect to spurious features) of the constructed subsampled dataset. To address this issue, we introduce a straightforward strategy called High Confidence Selection (HCS). The intuition is that although the spurious label of some examples in the minority groups may be misclassified, they are mostly close to the classification boundary, i.e., falling in the low-confidence region. Therefore, regarding the minority groups identified by DISK (i.e., the groups with smaller size), we only pick the examples with high-confidence predictions (parameterized by $> \alpha$ for some $\alpha > 0.5$, based on the predicted probability) while ignoring the low-confidence examples. Then, based on the smallest size of the predicted groups (after performing HCS), we will equally pick the same number of data points from all groups to form the subsampled dataset, which will be used for mitigating the subpopulation shift and enhance the generalization performance in the test domain.

## 4 EXPERIMENTS

In this section, we extensively evaluate DISK on a 2D synthetic dataset and five real-world image datasets, primarily addressing the following questions:

Q1. Can DISK accurately infer domains and effectively facilitate subpopulation shift mitigation?

Q2. If the inference is inaccurate, why is there inconsistency between DISK and human decisions in domain inference?

We also address the challenge posed by Lin et al. (2022) in inferring domains for CMNIST and MCOLOR in Appendix B.3.3 to demonstrate that DISK accurately infers domain information in heterogeneous data without facing the same difficulties as EIIL and LfF.

---

[1] Retraining the model is not mandatory and can also be done as suggested in Kirichenko et al. (2022); Izmailov et al. (2022), where a pre-trained model is employed, and only the last layer is fine-tuned.

## 4.1 SYNTHETIC 2D DATA

We begin with a synthetic 2D dataset to show how DISK partitions domains by learning from spurious correlations. This dataset comprises three sets: training ($\mathcal{D}^{tr}$), validation ($\mathcal{D}^{val}$), and test ($\mathcal{D}^{ts}$). These sets are generated by blending four two-dimensional Gaussian distributions, each with different means, equal variances, and zero correlation coefficients. Varying mixing probabilities across datasets induce subpopulation shift in $\mathcal{D}^{tr}$, $\mathcal{D}^{val}$, and $\mathcal{D}^{ts}$. The first dimension, $\mathbf{x}_1$, represents the spurious feature, while the second dimension, $\mathbf{x}_2$, is the invariant feature. More details about the synthetic data can be found in Appendix B.1. We trained a single-layer neural network, referred to as $f_{\text{vanilla}}$, on $\mathcal{D}^{tr}$ and visualized its decision boundary in the top row of Figure 3. We observed a significant accuracy gap between the training and test sets, with $f_{\text{vanilla}}$ aligning its decision boundary more closely with the vertical boundary determined by the spurious feature $\mathbf{x}_1$ rather than the horizontal boundary determined by the invariant feature $\mathbf{x}_2$. This indicates that $f_{\text{vanilla}}$ heavily relied on $\mathbf{x}_1$ for classification, resulting in poor test set generalization.

When $y^{val}$ is available, we used DISK to train $f_{\text{DISK}}$ with the same model architecture as $f_{\text{vanilla}}$, assigning distinct spurious labels to each instance, representing different domains. As shown in the second row of Figure 3, DISK indeed caused the decision boundary of $f_{\text{DISK}}$ to align more closely with the vertical boundary, leading to a more significant difference in prediction accuracy between the training and validation sets. Spurious labels and true labels divided the data space into four groups. We then applied a subsampling strategy to obtain an equal number of instances from each group, creating a balanced subsampled dataset. Subsequently, we trained the same single-layer neural network, denoted as $f_{\text{DISKS}}$, on this subsampled data and obtained its decision boundary and accuracy, as depicted in the third row of Figure 3. Compared to $f_{\text{vanilla}}$, the decision boundary of $f_{\text{DISK}}$ is noticeably more horizontal, and the test accuracy improved from 0.73 to 0.89, indicating reduced reliance on spurious features and enhanced model generalization. Additional experimental results without $y^{val}$ in Appendix B.1 yield similar outcomes.

## 4.2 REAL-WORLD DATA

To address Q1 in Section 4, we report the test prediction accuracy of DISK with Subsampling (abbreviated as DISKS) and baselines on five public real-world datasets, along with a metric to clarify domain inference effectiveness. To address Q2, we then conduct dataset-specific analysis based on the results from Q1. This analysis aims to explain the sources of discrepancies between the domain information inferred by DISK and by humans (oracle). Additionally, we showcase DISK's effectiveness when combined with other enhanced techniques, such as Mixup (abbreviated as DISKM), in Appendix B.3.4.

### 4.2.1 EXPERIMENTAL SETUP

**Datasets.** We consider image classification tasks with various spurious correlations. Specifically, the CMNIST dataset (Arjovsky et al., 2019) involves noisy digit recognition where digit colors (red or green) are spurious features linked to digit values. MNIST-FashionMNIST and MNIST-CIFAR (Shah et al., 2020; Kirichenko et al., 2022) are both synthetic datasets combining MNIST (spurious features) with FashionMNIST and CIFAR datasets, respectively. Additionally, we consider the Waterbirds dataset (Sagawa et al., 2019), which associates bird types with spurious background (water or land). Moreover, the CelebA dataset (Liu et al., 2015) focuses on hair color recognition, influenced by spurious gender-related features. More details of datasets are available in Appendix B.2.1.

**Baselines.** As discussed in Section 2, existing domain inference methods have limitations, including instability (as seen in EIIL and LfF) and data-specific applicability (as seen in ZIN and DISC), which restricts their usefulness as reliable baselines. Therefore, except for vanilla ERM (Vapnik, 1991), we consider domain generalization models that directly leverage oracle domain information, including IRM (Arjovsky et al., 2019), GroupDRO (Sagawa et al., 2019), LISA (Yao et al., 2022), and DFR (Kirichenko et al., 2022) as our baseline methods. Importantly, DFR uses oracle domain information for subsampling which makes comparing DISKS to DFR a direct validation of DISKS' effectiveness. Especially when the oracle domain information accurately represents spurious features, DFR sets the upper limit for DISKS' performance.

Table 1: Average and worst accuracy comparison (%). DISKS outperforms ERM, effectively mitigating the subpopulation shift issue without relying on domain information. The experimental results for LISA, GroupDRO, and IRM are directly sourced from (Yao et al., 2022).

| Method | Domain Info | CMNIST | | MNIST FashionMNIST | | MNIST CIFAR | | WaterBirds | | CelebA | |
|---|---|---|---|---|---|---|---|---|---|---|---|
| | | Avg. | Worst | Avg. | Worst | Avg. | Worst | Avg. | Worst | Avg. | Worst |
| ERM | ✗ | $37.8_{\pm1.1}$ | $32.4_{\pm1.2}$ | $71.1_{\pm2.0}$ | $69.6_{\pm2.0}$ | $9.8_{\pm0.3}$ | $9.6_{\pm6.0}$ | $63.4_{\pm4.0}$ | $34.4_{\pm8.2}$ | $94.7_{\pm0.8}$ | $38.0_{\pm3.4}$ |
| IRM | ✓ | $72.2_{\pm1.1}$ | $70.3_{\pm0.8}$ | - | - | - | - | $87.5_{\pm0.7}$ | $75.6_{\pm3.1}$ | $94.0_{\pm0.4}$ | $77.8_{\pm3.9}$ |
| GroupDRO | ✓ | $72.3_{\pm1.2}$ | $68.6_{\pm0.8}$ | - | - | - | - | $91.8_{\pm0.3}$ | $90.6_{\pm1.1}$ | $91.2_{\pm0.4}$ | $87.2_{\pm1.6}$ |
| LISA | ✓ | $74.0_{\pm0.1}$ | $73.3_{\pm0.2}$ | $92.9_{\pm0.7}$ | $92.6_{\pm0.8}$ | - | - | $91.8_{\pm0.3}$ | $89.2_{\pm0.6}$ | $92.4_{\pm0.4}$ | $89.3_{\pm1.1}$ |
| DFR | ✓ | $64.1_{\pm1.5}$ | $67.9_{\pm1.8}$ | $95.8_{\pm0.4}$ | $95.5_{\pm0.5}$ | $69.3_{\pm0.9}$ | $70.0_{\pm1.2}$ | $79.3_{\pm2.2}$ | $78.2_{\pm3.6}$ | $91.1_{\pm0.1}$ | $85.0_{\pm2.1}$ |
| DISKS $_{w/\ y^{val}}$ | ✗ | $65.1_{\pm1.7}$ | $67.6_{\pm2.0}$ | $92.3_{\pm0.8}$ | $92.6_{\pm0.9}$ | $69.0_{\pm0.4}$ | $69.2_{\pm0.6}$ | $91.1_{\pm1.4}$ | $85.5_{\pm3.0}$ | $88.8_{\pm0.3}$ | $64.8_{\pm1.3}$ |
| DISKS $_{w/o\ y^{val}}$ | ✗ | $62.5_{\pm4.4}$ | $65.5_{\pm3.0}$ | $91.8_{\pm2.8}$ | $93.0_{\pm2.7}$ | $68.1_{\pm1.2}$ | $68.4_{\pm1.2}$ | $80.8_{\pm1.5}$ | $81.1_{\pm0.4}$ | $87.9_{\pm0.4}$ | $63.0_{\pm4.6}$ |

**Model Training.** We adopt the neural network architectures and hyperparameters from Yao et al. (2022), employing ResNet (He et al., 2016) models as the feature extractor $\Phi(\mathbf{x})$, which updates during training. We conduct three repetitions for each model, reporting mean and standard deviation following Yao et al. (2022). Further training specifics are in Appendix B.2.2.

**Evaluation.** For all datasets, we evaluate average-group and worst-group prediction accuracies. Additionally, we introduce a new metric, Minority Domain Inference Precision ($P_{\mathrm{M}}$), to showcase DISK's ability to identify challenging minority groups. This metric is defined as follows:

**Definition 2** (Minority Domain Inference Precision). *The index set of instances from minority groups inferred by oracle domain information is denoted as $I_{\mathrm{oracle}}$, while those inferred by DISK are denoted as $I_{\mathrm{DISK}}$. We define Minority Domain Inference Precision ($P_{\mathrm{M}}$) as $P_{\mathrm{M}} = \frac{|I_{\mathrm{oracle}} \cap I_{\mathrm{DISK}}|}{|I_{\mathrm{DISK}}|}$.*

### 4.2.2 RESULTS

**Can DISK accurately infer domains and effectively facilitate subpopulation shift mitigation?**

Table 1 presents a performance comparison of DISKS with baseline models across five datasets. DISKS consistently achieves significantly improved average and worst accuracy compared to traditional ERM, regardless of the availability of $y^{val}$. On some datasets, DISKS nearly matches or even surpasses the performance of models like DFR, which utilize oracle domain information directly. We also observed that in most datasets, DISKS without $y^{val}$ exhibits slightly lower average/worst accuracy compared to those with $y^{val}$. This might be attributed to the challenges associated with optimizing the KL-divergence of high-dimensional inputs (i.e., the representation $\mathbf{z}$) by DISKS without $y^{val}$ (Belghazi et al., 2018). Furthermore, it's worth noting that DISK's performance improvements are less pronounced on the CelebA dataset, possibly due to the limited introduction of additional domain information resulting from the close alignment between the training and validation set distributions. Table 2 further summarizes DISK's accuracy in identifying minority groups. Across all

Table 2: Average $P_{\mathrm{M}}$. DISK maintains high precision in recognizing minority groups.

| Method | CMNIST | MNIST FashionMNIST | MNIST CIFAR | WaterBirds | CelebA |
|---|---|---|---|---|---|
| DISK $_{w/\ y^{val}}$ | $99.1_{\pm0.2}$ | $91.8_{\pm1.7}$ | $99.2_{\pm0.1}$ | $70.3_{\pm3.1}$ | $71.3_{\pm2.0}$ |
| DISK $_{w/o\ y^{val}}$ | $98.5_{\pm0.5}$ | $94.9_{\pm2.4}$ | $98.3_{\pm1.3}$ | $76.7_{\pm3.0}$ | $69.7_{\pm2.2}$ |

datasets, the average precision ($P_{\mathrm{M}}$) for minority groups consistently exceeds 70%, and in some cases, even reaches as high as 90%. Furthermore, Figure 4 visualizes the distribution of spurious features in the subsampled data. DISK enables the creation of subsampled data with more balanced spurious feature distributions within each class, facilitating models in learning invariant features and improving generalization. In Appendix B.3.2, we also show the ablation experiment results to demonstrate that DISK, without performing HCS, still accurately infers domains and effectively mitigates subpopulation shift.

**Why is there inconsistency between DISK and human decisions in domain inference?**

We noticed an interesting phenomenon in Table 1 and Table 2: in the Waterbirds dataset, DISK only achieves around 70% precision on minority groups, yet its performance approaches or even exceeds that of DFR. To explain this, Figure 5 displays ten random instances from the minority groups identified by DISK. We observe two main categories of misclassified images: (1) Land images with

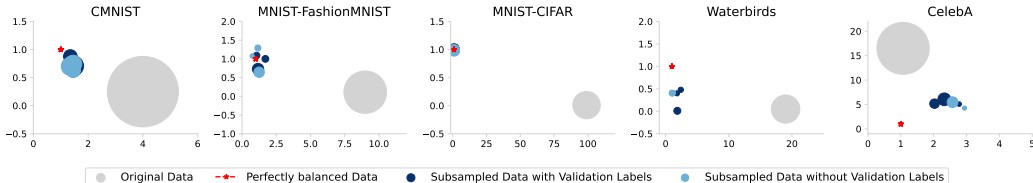

Figure 4: Subsampled data visualization. The x-axis shows the spurious feature ratio $\frac{\mathbb{P}(\mathbf{x}_s=0|y=0)}{\mathbb{P}(\mathbf{x}_s=1|y=0)}$ in class 0 , the y-axis shows the ratio $\frac{\mathbb{P}(\mathbf{x}_s=0|y=1)}{\mathbb{P}(\mathbf{x}_s=1|y=1)}$ in class 1 , and bubble size represents sample size. Compared to the original data, subsampled data by DISK exhibits improved balance of spurious features within each class, approaching closer to the perfectly balanced data (closer to (1,1)).

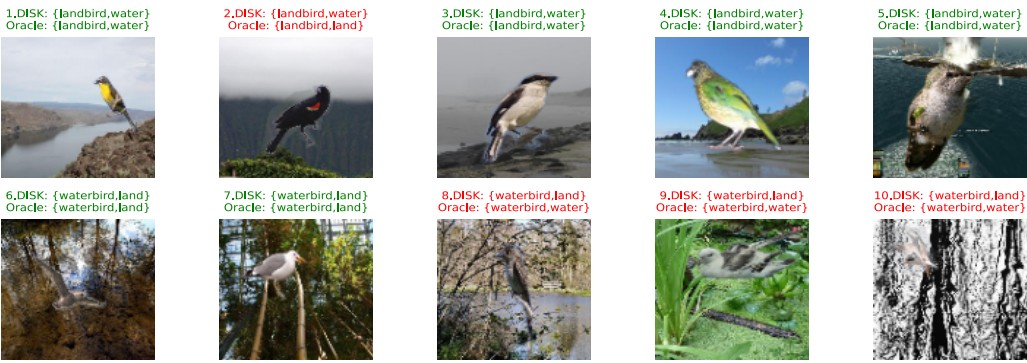

Figure 5: Comparing DISK-inferred and oracle minority groups. Each image has labels from both DISK and the oracle, with "waterbird/landbird" as true labels and "water/land" as spurious (domain) labels. Red highlights the DISK and oracle mismatch in domain classification.

typical water features, like extensive blue regions (Figure 2), often lead DISK to misclassify land as water. (2) Water images with typical land features, such as abundant tree branches (Figure 8), ponds with lush green vegetation (Figure 9), or large tree reflections (Figure 10), frequently cause DISK to misclassify water as land. Specifically, in Figure 2 of Figure 5, DISK misclassifies it as water when it is actually land. We notice that it shares nearly identical background structures with Figures 1, 3, and 4 in Figure 5: vast blue areas (ocean or sky) and yellow/green land. It's reasonable for DISK to group them due to similar backgrounds. Unlike Figures 8 and 9 in Figure 5, which were misclassified as land because their main content directly includes many land elements, such as green foliage and tree branches, Figure 10 is classified as land by DISK, despite its water background, due to the abundance of vertical linear structures resembling typical land features (tree branches).

Appendix B.3.5 includes additional visualizations that support our conclusion: for the Waterbirds dataset, DISK achieves more coherent domain partitioning than human decisions by grouping spurious features (backgrounds) with similar underlying patterns into the same category. For instance, DISK identifies similarities between tree branches and water reflections, recognizes scenes resembling vast blue skies and oceans, and groups them accordingly. And DISK provides domain inference that is entirely based on the neural network perspective. This maintains perspective consistency with subsequent neural networks trained on datasets without domain interference, thereby creating an end-to-end process that can mitigate the adverse effects arising from differences in cognition between humans and neural networks.

## 5 CONCLUSION

To address subpopulation shift without domain information, we introduce DISK, a novel method for inferring domain labels. We evaluate its effectiveness across various datasets, emphasizing its domain inference capabilities. In the WaterBirds dataset, DISK outperforms human decisions, suggesting its potential in capturing the essence of data and demonstrating DISK's value even when domain information is available. However, limitations become evident in datasets like CelebA, where closely aligned distributions challenge the recognition of spurious correlations.

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

# A PROOF DETAILS

## A.1 PROOF OF THE TRANSFORMATION OF $I(y^{tr}; y^{tr}_{s,\mathbf{w}})$

Given $y^{tr}$, the maximization of mutual information $I(y^{tr}; y^{tr}_{s,\mathbf{w}})$ can be transformed into the minimization of cross-entropy $H(y^{tr}, y^{tr}_{s,\mathbf{w}})$.

To demonstrate this assertion, we establish the following lemma:

**Lemma 1.** *Given $y^{tr}$, the correlation term is lower bounded by the difference between the entropy of $y^{tr}$ and the cross-entropy between $y^{tr}$ and $y^{tr}_{s,\mathbf{w}}$:*

$$I(y^{tr}; y^{tr}_{s,\mathbf{w}}) \geq H(y^{tr}) - H(y^{tr}, y^{tr}_{s,\mathbf{w}}).$$

*Proof.* First, we expand the mutual information term:

$$I(y^{tr}; y^{tr}_{s,\mathbf{w}}) = H(y^{tr}) - H(y^{tr}|y^{tr}_{s,\mathbf{w}})$$

Since, the cross-entropy can be expanded as:

$$
\begin{aligned}
H(y^{tr}, y^{tr}_{s,\mathbf{w}}) &= -\sum \mathbb{P}(y^{tr}) \log \mathbb{P}(y^{tr}_{s,\mathbf{w}}) \\
&= \sum \mathbb{P}(y^{tr}) \log \frac{\mathbb{P}(y^{tr})}{\mathbb{P}(y^{tr}_{s,\mathbf{w}})} - \sum \mathbb{P}(y^{tr}) \log \mathbb{P}(y^{tr}) \\
&= D_{\mathbf{KL}}(y^{tr}||y^{tr}_{s,\mathbf{w}}) + H(y^{tr}) \\
&\geq D_{\mathbf{KL}}(y^{tr}||y^{tr}_{s,\mathbf{w}}) + H(y^{tr}|y^{tr}_{s,\mathbf{w}}) \\
&\geq H(y^{tr}|y^{tr}_{s,\mathbf{w}})
\end{aligned}
$$

Then we have,

$$
\begin{aligned}
I(y^{tr}; y^{tr}_{s,\mathbf{w}}) &= H(y^{tr}) - H(y|y^{tr}_{s,\mathbf{w}}) \\
&\geq H(y^{tr}) - H(y^{tr}, y^{tr}_{s,\mathbf{w}})
\end{aligned}
$$

$\square$

According to Lemma 1, when $y^{tr}$ is given, maximizing the mutual information $I(y^{tr}; y^{tr}_{s,\mathbf{w}})$ can be equivalently transformed into minimizing $H(y^{tr}, y^{tr}_{s,\mathbf{w}})$, which is more computationally tractable in practice.

## A.2 PROOF OF THEOREM 1

To prove Theorem 1, we first show the causal structure between $y_s$, $\mathbf{z}$ and $y$. Definition 1 describes the following causal relationships: $y_s \leftarrow \mathbf{z}_s$ and $\mathbf{z}_v \rightarrow y$. This relationship is reasonable and further complements the research by Li et al. (2022). In their work, they assume $D \leftarrow \mathbf{z} \rightarrow y$, where the variable $D$ represents the domain, which corresponds to the spurious labels $y_s$. Here, $\mathbf{z} = (\mathbf{z}_s, \mathbf{z}_v)$, and we provide a detailed description of how $\mathbf{z}$ determines both the domain label and the true label through $\mathbf{z}_s$ and $\mathbf{z}_v$. The fork causal structure (Lagnado & Sloman, 2019) between $y_s$, $\mathbf{z}$ and $y$ exhibits the following properties:

**Property 1.** *The fork causal relationship $y_s \leftarrow \mathbf{z} \rightarrow y$ adheres to the following properties:*

1. *$y_s \not\perp y$ means the true label $y$ and the spurious label $y_s$ are dependent.*

2. *$y \perp y_s \,|\, \mathbf{z}$ means given the representation $\mathbf{z}$, the true label $y$ and the spurious label $y_s$ are conditionally independent.*

Then, we establish the following two lemmas:

**Lemma 2.** *Given representations $\mathbf{z}^{tr}$ and $\mathbf{z}^{val}$, maximizing the spurious term is equivalent to maximizing the following expression:*

$$
\begin{aligned}
&\max_{\mathbf{w}} \mathrm{KL}(\mathbb{P}(y^{tr}|y^{tr}_{s,\mathbf{w}})||\mathbb{P}(y^{val}|y^{val}_{s,\mathbf{w}})) \\
&\Leftrightarrow \max_{\mathbf{w}} \mathrm{KL}(\mathbb{P}(y^{tr}, \mathbf{z}^{tr}|y^{tr}_{s,\mathbf{w}})||\mathbb{P}(y^{val}, \mathbf{z}^{val}|y^{val}_{s,\mathbf{w}}))
\end{aligned}
\tag{7}
$$

*Proof.* According to the properties of KL divergence (Cover, 1999), we have the following:

$$
\begin{aligned}
&\mathrm{KL}(\mathbb{P}(y^{tr}, \mathbf{z}^{tr}|y^{tr}_{s,\mathbf{w}})||\mathbb{P}(y^{val}, \mathbf{z}^{val}|y^{val}_{s,\mathbf{w}})) - \mathrm{KL}(\mathbb{P}(\mathbf{z}^{tr}|y^{tr}, y^{tr}_{s,\mathbf{w}})||\mathbb{P}(\mathbf{z}^{val}|y^{val}, y^{val}_{s,\mathbf{w}})) \\
&= \mathbb{E}_{(y^{tr},\mathbf{z}^{tr},y^{tr}_{s,\mathbf{w}})}[\log \frac{\mathbb{P}(y^{tr}, \mathbf{z}^{tr}|y^{tr}_{s,\mathbf{w}})}{\mathbb{P}(y^{val}, \mathbf{z}^{val}|y^{val}_{s,\mathbf{w}})}] - \mathbb{E}_{(y^{tr},\mathbf{z}^{tr},y^{tr}_{s,\mathbf{w}})}[\log \frac{\mathbb{P}(\mathbf{z}^{tr}|y^{tr}, y^{tr}_{s,\mathbf{w}})}{\mathbb{P}(\mathbf{z}^{val}|y^{val}, y^{val}_{s,\mathbf{w}})}] \\
&= \mathbb{E}_{(y^{tr},\mathbf{z}^{tr},y^{tr}_{s,\mathbf{w}})}[\log \frac{\mathbb{P}(y^{tr}|y^{tr}_{s,\mathbf{w}})}{\mathbb{P}(y^{val}|y^{val}_{s,\mathbf{w}})}] \\
&= \mathbb{E}_{(Y^{tr},y^{tr}_{s,\mathbf{w}})}[\log \frac{\mathbb{P}(Y^{tr}|y^{tr}_{s,\mathbf{w}})}{\mathbb{P}(Y^{val}|y^{val}_{s,\mathbf{w}})}] \\
&= \mathrm{KL}(\mathbb{P}(y^{tr}|y^{tr}_{s,\mathbf{w}})||\mathbb{P}(y^{val}|y^{val}_{s,\mathbf{w}}))
\end{aligned}
\tag{8}
$$

Term $\mathrm{KL}(\mathbb{P}(\mathbf{z}^{tr}|y^{tr}, y^{tr}_{s,\mathbf{w}})||\mathbb{P}(\mathbf{z}^{val}|y^{val}, y^{val}_{s,\mathbf{w}}))$ captures the similarity between the groups in the training and validation sets. According to the definition of subpopulation shift in Section 3.1, we have $\mathbb{P}(\mathbf{z}^{tr}|y^{tr}, y^{tr}_{s,\mathbf{w}}) = \mathbb{P}(\mathbf{z}^{val}|y^{val}, y^{val}_{s,\mathbf{w}})$. Therefore,

$$
\begin{aligned}
\mathrm{KL}(\mathbb{P}(\mathbf{z}^{tr}|y^{tr}, y^{tr}_{s,\mathbf{w}})||\mathbb{P}(\mathbf{z}^{val}|y^{val}, y^{val}_{s,\mathbf{w}})) &= \sum \mathbb{P}(y^{tr}, y^{tr}_{s,\mathbf{w}}) \sum \mathbb{P}(\mathbf{z}^{tr}|y^{tr}, y^{tr}_{s,\mathbf{w}}) \log \frac{\mathbb{P}(\mathbf{z}^{tr}|y^{tr}, y^{tr}_{s,\mathbf{w}})}{\mathbb{P}(\mathbf{z}^{val}|y^{val}, y^{val}_{s,\mathbf{w}})} \\
&= \sum_k r^{tr}_k \sum P_k \log \frac{\mathbb{P}_k}{\mathbb{P}_k} = 0
\end{aligned}
\tag{9}
$$

Consequently, maximizing the spurious term $\mathrm{KL}(\mathbb{P}(y^{tr}|y^{tr}_{s,\mathbf{w}})||\mathbb{P}(y^{val}|y^{val}_{s,\mathbf{w}}))$ is tantamount to maximizing $\mathrm{KL}(\mathbb{P}(y^{tr}, \mathbf{z}^{tr}|y^{tr}_{s,\mathbf{w}})||\mathbb{P}(y^{val}, \mathbf{z}^{val}|y^{val}_{s,\mathbf{w}}))$. $\qquad\square$

**Lemma 3.** *Given representations $\mathbf{z}^{tr}$ and $\mathbf{z}^{val}$, the equivalence of the spurious term is lower bounded by the following expression:*

$$
\mathrm{KL}(\mathbb{P}(y^{tr}, \mathbf{z}^{tr}|y^{tr}_{s,\mathbf{w}})||\mathbb{P}(y^{val}, \mathbf{z}^{val}|y^{val}_{s,\mathbf{w}})) \geq \mathrm{KL}(\mathbb{P}(\mathbf{z}^{tr}|y^{tr}_{s,\mathbf{w}})||\mathbb{P}(\mathbf{z}^{val}|y^{val}_{s,\mathbf{w}}))
\tag{10}
$$

*The tighter the bound is as $\mathrm{KL}(\mathbb{P}(y^{tr}|\mathbf{z}^{tr})||\mathbb{P}(y^{val}|\mathbf{z}^{val}))$ becomes smaller.*

*Proof.* Expanding the objective term, we have:

$$
\begin{aligned}
&\mathrm{KL}(\mathbb{P}(y^{tr}, \mathbf{z}^{tr}|y^{tr}_{s,\mathbf{w}})||\mathbb{P}(y^{val}, \mathbf{z}^{val}|y^{val}_{s,\mathbf{w}})) \\
&= \mathbb{E}_{(y^{tr},\mathbf{z}^{tr},y^{tr}_{s,\mathbf{w}})}[\log \frac{\mathbb{P}(y^{tr}, \mathbf{z}^{tr}|y^{tr}_{s,\mathbf{w}})}{\mathbb{P}(y^{val}, \mathbf{z}^{val}|y^{val}_{s,\mathbf{w}})}] \\
&= \mathbb{E}_{(y^{tr},\mathbf{z}^{tr},y^{tr}_{s,\mathbf{w}})}[\log \frac{\mathbb{P}(y^{tr}, \mathbf{z}^{tr}, y^{tr}_{s,\mathbf{w}})}{\mathbb{P}(y^{val}, \mathbf{z}^{val}, y^{val}_{s,\mathbf{w}})}] - \mathbb{E}_{y^{tr}_{s,\mathbf{w}}}[\log \frac{\mathbb{P}(y^{tr}_{s,\mathbf{w}})}{\mathbb{P}(y^{val}_{s,\mathbf{w}})}] \\
&\stackrel{(a)}{=} \mathbb{E}_{(y^{tr},\mathbf{z}^{tr},y^{tr}_{s,\mathbf{w}})}[\log \frac{\mathbb{P}(y^{tr}|\mathbf{z}^{tr})\mathbb{P}(\mathbf{z}^{tr}, y^{tr}_{s,\mathbf{w}})}{\mathbb{P}(y^{val}|\mathbf{z}^{val})\mathbb{P}(\mathbf{z}^{val}, y^{val}_{s,\mathbf{w}})}] - \mathrm{KL}(\mathbb{P}(y^{tr}_{s,\mathbf{w}})||\mathbb{P}(y^{val}_{s,\mathbf{w}})) \\
&= \mathbb{E}_{(y^{tr},\mathbf{z}^{tr},y^{tr}_{s,\mathbf{w}})}[\log \frac{\mathbb{P}(y^{tr}|\mathbf{z}^{tr})}{\mathbb{P}(y^{val}|\mathbf{z}^{val})}] + \mathbb{E}_{(y^{tr},\mathbf{z}^{tr},y^{tr}_{s,\mathbf{w}})}[\log \frac{\mathbb{P}(\mathbf{z}^{tr}, y^{tr}_{s,\mathbf{w}})}{\mathbb{P}(\mathbf{z}^{val}, y^{val}_{s,\mathbf{w}})}] - \mathrm{KL}(\mathbb{P}(y^{tr}_{s,\mathbf{w}})||\mathbb{P}(y^{val}_{s,\mathbf{w}})) \\
&= \mathrm{KL}(\mathbb{P}(y^{tr}|\mathbf{z}^{tr})||\mathbb{P}(y^{val}|\mathbf{z}^{val})) + \mathrm{KL}(\mathbb{P}(\mathbf{z}^{tr}|y^{tr}_{s,\mathbf{w}})||\mathbb{P}(\mathbf{z}^{val}|y^{val}_{s,\mathbf{w}})) \\
&\geq \mathrm{KL}(\mathbb{P}(\mathbf{z}^{tr}|y^{tr}_{s,\mathbf{w}})||\mathbb{P}(\mathbf{z}^{val}|y^{val}_{s,\mathbf{w}}))
\end{aligned}
\tag{11}
$$

(a) achieves by the Property 1 where we have:

$$\mathbb{P}(y, \mathbf{z}, y_s) = \mathbb{P}(y|\mathbf{z})\mathbb{P}(y_s|\mathbf{z})\mathbb{P}(\mathbf{z}) = \mathbb{P}(y|\mathbf{z})\mathbb{P}(y_s, \mathbf{z}) \tag{12}$$

By Equation 11, we prove the lower bound of $\mathrm{KL}(\mathbb{P}(y^{tr}, \mathbf{z}^{tr}|y_{s,\mathbf{w}}^{tr})||\mathbb{P}(y^{val}, \mathbf{z}^{val}|y_{s,\mathbf{w}}^{val}))$ is $\mathrm{KL}(\mathbb{P}(\mathbf{z}^{tr}|y_{s,\mathbf{w}}^{tr})||\mathbb{P}(\mathbf{z}^{val}|y_{s,\mathbf{w}}^{val}))$ and the tighter the bound is as $\mathrm{KL}(\mathbb{P}(y^{tr}|\mathbf{z}^{tr})||\mathbb{P}(y^{val}|\mathbf{z}^{val}))$ becomes smaller.

$\square$

**Theorem 2.** *(Restatement of Theorem 1) Given representations $\mathbf{z}^{tr}$ and $\mathbf{z}^{val}$, the spurious term is lower bounded by the following expression as:*

$$\mathrm{KL}(\mathbb{P}(y^{tr}|y_{s,\mathbf{w}}^{tr})||\mathbb{P}(y^{val}|y_{s,\mathbf{w}}^{val})) \geq \mathrm{KL}(\mathbb{P}(\mathbf{z}^{tr}|y_{s,\mathbf{w}}^{tr})||\mathbb{P}(\mathbf{z}^{val}|y_{s,\mathbf{w}}^{val})) \tag{13}$$

*Proof.* Based on Lemma 2 and 3, we can directly deduce:

$$\mathrm{KL}(\mathbb{P}(y^{tr}|y_{s,\mathbf{w}}^{tr})||\mathbb{P}(y^{val}|y_{s,\mathbf{w}}^{val})) \geq \mathrm{KL}(\mathbb{P}(\mathbf{z}^{tr}|y_{s,\mathbf{w}}^{tr})||\mathbb{P}(\mathbf{z}^{val}|y_{s,\mathbf{w}}^{val})) \tag{14}$$

$\square$

# B EXPERIMENTS

## B.1 SYNTHETIC TOY DATA

Consider a 2D synthetic dataset with the following distribution:

**Example 1.** *(Synthetic 2D data) Let $\mathbf{x} = (\mathbf{x}_1, \mathbf{x}_2) \in \mathbb{R}^2$ represent 2-dimensional features, with the spurious feature $X_1$ and the invariant feature $\mathbf{x}_2$, and $y \in \mathbb{R}^1$ denoting labels. The synthetic data comprises four groups (domains), namely $G_1$, $G_2$, $G_3$, and $G_4$. The distributions and sample sizes in the training, validation, and test sets for each group are as follows:*

$$\begin{cases} G_1 : (\mathbf{x}_1, \mathbf{x}_2) \sim \mathcal{N}\left(\begin{bmatrix} 4 \\ 5 \end{bmatrix}, \begin{bmatrix} 1 & 0 \\ 0 & 1 \end{bmatrix}\right); Y = 0; (N^{tr}, N^{val}, N^{ts}) = (3900, 854, 3000) \\ G_2 : (\mathbf{x}_1, \mathbf{x}_2) \sim \mathcal{N}\left(\begin{bmatrix} 4 \\ 8 \end{bmatrix}, \begin{bmatrix} 1 & 0 \\ 0 & 1 \end{bmatrix}\right); Y = 1; (N^{tr}, N^{val}, N^{ts}) = (100, 287, 3000) \\ G_3 : (\mathbf{x}_1, \mathbf{x}_2) \sim \mathcal{N}\left(\begin{bmatrix} 8 \\ 8 \end{bmatrix}, \begin{bmatrix} 1 & 0 \\ 0 & 1 \end{bmatrix}\right); Y = 1; (N^{tr}, N^{val}, N^{ts}) = (3900, 18, 3000) \\ G_4 : (\mathbf{x}_1, \mathbf{x}_2) \sim \mathcal{N}\left(\begin{bmatrix} 8 \\ 5 \end{bmatrix}, \begin{bmatrix} 1 & 0 \\ 0 & 1 \end{bmatrix}\right); Y = 0; (N^{tr}, N^{val}, N^{ts}) = (100, 828, 3000) \end{cases} \tag{15}$$

*The varying sample sizes in groups $G_1$, $G_2$, $G_3$, and $G_4$ indicate subpopulation shift across the training, validation, and test sets.*

Figure 6 visualizes the synthetic data and annotates the centers of the four groups. Subpopulation shift can be clearly observed in the training, validation, and test datasets. Specifically, all three datasets contain four groups: $G_1$, $G_2$, $G_3$, and $G_4$. However, the proportions of these groups vary significantly across the datasets. Such distribution shift leads to challenges in generalization, as models trained on the training set may struggle to perform well on the validation and test sets, resulting in poor overall generalization performance.

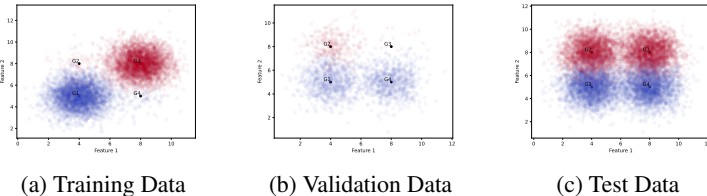

(a) Training Data      (b) Validation Data      (c) Test Data

Figure 6: Visualization of Synthetic Data. The centers of the four groups $G_1$, $G_2$, $G_3$, and $G_4$ are labeled. We can observe the subpopulation shift phenomenon in the training, validation, and test datasets.

The second row of Figure 7 visualizes the decision boundary of $f_{\text{DISK}}$ without access to $y^{\text{val}}$. Compared to $f_{\text{vanilla}}$, a decision boundary that aligns more closely with the vertical decision driven by the spurious feature x1 is observed. This implies that $f$DISK without $y^{\text{val}}$ has learned more spurious features, resulting in poorer generalization on the test data.

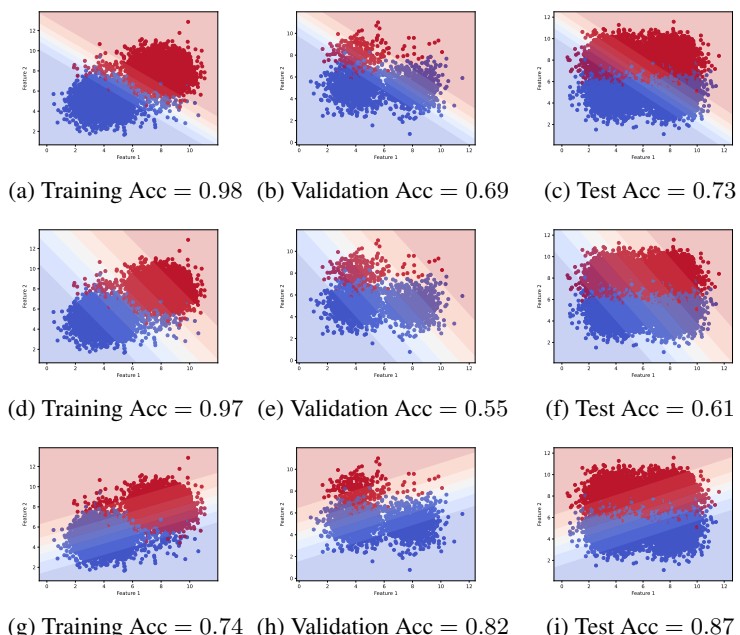

(a) Training Acc = 0.98   (b) Validation Acc = 0.69   (c) Test Acc = 0.73

(d) Training Acc = 0.97   (e) Validation Acc = 0.55   (f) Test Acc = 0.61

(g) Training Acc = 0.74   (h) Validation Acc = 0.82   (i) Test Acc = 0.87

Figure 7: (a-c) Decision Boundary and Prediction Accuracy for $f_{\text{vanilla}}$. (d-f) DISK without Accessible $Y^{val}$: Decision Boundary and Prediction Accuracy for $f_{\text{DISK}}$. (g-i) DISK with Subsampling: Decision Boundary and Prediction Accuracy for $f_{\text{DISKS}}$.

The third row of Figure 7 visualizes $f_{\text{DISKS}}$, which is furthermore trained based on the domain information inferred by DISK using the Subsampling strategy. Compared to $f_{\text{vanilla}}$, $f_{\text{DISKS}}$ exhibits a more horizontal decision boundary and higher test prediction accuracy. This indicates that DISK can effectively infer domain information to help mitigate subpopulation shift.

## B.2 REAL DATA

### B.2.1 DATASET DETAILS

**CMNIST**(Arjovsky et al., 2019) is a noisy digit recognition task. The binary feature (green and red), referred to as color, serves as a spurious feature, while the binary feature (digit contours) acts as the invariant feature. The CMNIST dataset involves two classes, where class 0 corresponds to the original digits (0,1,2,3,4), and class 1 represents digits (5,6,7,8,9). Following the approach

recommended in Yao et al. (2022), we construct a training set with a sample size of 30,000. In class 0, the ratio of red to green samples is set at 8:2, while in class 1, it is set at 2:8. For the validation set consisting of 10,000 samples, the proportion of green to red samples is equal at 1:1 for all classes. The test set, containing 20,000 samples, features a proportion of green to red samples at 1:9 in class 0 and 9:1 in class 1. Additionally, label flipping is applied with a probability of 0.25.

**MNIST-FashionMNIST** are synthetic datasets derived from the Dominoes datasets (Shah et al., 2020; Kirichenko et al., 2022). MNIST-FashionMNIST is generated by combining the MNIST dataset with the FashionMNIST dataset. In these datasets, each image is divided into two halves: the top half displays MNIST digits from classes 0, 1, while the bottom half showcases Fashion-MNIST images from classes coat, dress. For MNIST-FashionMNIST, in the training set with 10,825 samples, in the class Dress, the ratio of digit 0 to digit 1 samples is set at 9:1, while in the class Coat, it is set at 1:9. In the validation set with 1,175 samples, in the class Dress, the ratio of digit 0 to digit 1 samples is approximately set at 7:3, while in the class Coat, it is approximately set at 3:7. In the test set with 2,000 samples, 95% of the samples in the Dress class are associated with the digit 1, while 95% of the samples in the Coat class are associated with the digit 0.

**MNIST-CIFAR** are also synthetic datasets created by combining the MNIST dataset with the CIFAR datasets (Shah et al., 2020; Kirichenko et al., 2022). To increase the diversity of the data, we included all samples from CIFAR-10, rather than just the samples car, truck as included in Kirichenko et al. (2022). First, the labels of CIFAR-10 were binarized, where samples originally labeled as airplane, automobile, bird, cat, or deer were relabeled as 0, and samples originally labeled as dog, frog, horse, ship, or truck were relabeled as 1. In these datasets, each image is divided into two halves: the top half displays MNIST digits from classes 0, 1, while the bottom half showcases CIFAR-10 images from the new classes 0, 1. MNIST-CIFAR exhibits more extreme distributions of spurious features (MNIST) and invariant features (CIFAR-10). For the training set with 4,500 samples, 99% of samples labeled as 0 are associated with digit 0, and 99% of samples labeled as 1 are associated with digit 1. In the validation set with 5,000 samples, 70% of samples labeled as 0 are associated with digit 0, and 70% of samples labeled as 1 are associated with digit 1. In the test set with 10,000 samples, only 10% of samples labeled as 0 are associated with digit 0, and 10% of samples labeled as 1 are associated with digit 1.

**Waterbirds** aims to classify bird images as either waterbirds or landbirds, with each bird image falsely associated with either a water or land background. Waterbirds is a synthetic dataset where each image is generated by combining bird images sampled from the CUB dataset (Wah et al., 2011) with backgrounds selected from the Places dataset (Zhou et al., 2017). We directly load the Waterbirds dataset using the Wilds library in PyTorch (Koh et al., 2021). The dataset consists of a total of 4,795 training samples, with only 56 samples labeled as waterbirds on land and 184 samples labeled as landbirds on water. The remaining training data includes 3,498 samples from landbirds on land and 1,057 samples from waterbirds on water.

**CelebA** (Liu et al., 2015; Sagawa et al., 2019) is a hair-color prediction task, similar to the study conducted by (Yao et al., 2022), and follows the data preprocessing procedure outlined in (Sagawa et al., 2019). Given facial images of celebrities as input, the task is to identify their hair color as either blond or non-blond. This labeling is spuriously correlated with gender, which can be either male or female. In the training set, there are 71,629 instances (44%) of females with dark hair, 66,874 instances (41%) of non-blond males, 22,880 instances (14%) of blond females, and 1,387 instances (1%) of blond males. In the validation set, there are 8535 instances (43%) of females with dark hair, 8276 instances (42%) of non-blond males, 2874 instances (14%) of blond females, and 182 instances (1%) of blond males.

### B.2.2 TRAINING DETAILS

We use a pre-trained ResNet model (He et al., 2016) for image data. For Subsampling, assuming a minimum sample size of $T$ for each domain after DISK partitioning, we sample $T$ samples from each domain to form a subsampled dataset. Each method is repeated three times with random seeds 0, 1, and 2. Detailed parameters used in the experiments are shown in Table 3.

Table 3: Hyperparameter settings for Different Datasets. Parameters inside parentheses indicate differences between cases when DISK $y^{val}$ is not available and when $y^{val}$ is available. Parameters inside parentheses correspond to DISK with $y^{val}$ not available.

| | Hyperparameters | CMNIST | MNIST FashionMNIST | MNIST CIFAR | Waterbirds | CelebA |
|---|---|---|---|---|---|---|
| $f_{\text{vanilla}}$ | Learning rate | 1e-3 | 1e-2 | 1e-2 | 1e-3 | 1e-3 |
| | Weight decay | 1e-4 | 1e-3 | 1e-3 | 1e-4 | 1e-4 |
| | Architecture | ResNet50 | ResNet18 | ResNet18 | ResNet18 | ResNet50 |
| | Epoch | 300 | 300 | 300 | 300 | 50 |
| $f_{\text{DISK}}^{spurious}$ | Learning rate | 1e-5 | 1e-5 | 1e-5 | 1e-5 | 1e-5 |
| | Momentum | 0.9 | 0.9 | 0.9 | 0.9 | 0.9 |
| | Architecture | 1-layer NN | 1-layer NN | 1-layer NN | 1-layer NN | 1-layer NN |
| | Epoch | 100 (50) | 500 (500) | 300 (300) | 200 (100) | 180 (180) |
| $f_{\text{DISK}}^{correlation}$ | Learning rate | 1e-5 | 1e-4 | 1e-5 | 1e-5 | 1e-5 |
| | Momentum | 0.9 | 0.9 | 0.9 | 0.9 | 0.9 |
| | Architecture | 1-layer NN | 1-layer NN | 1-layer NN | 1-layer NN | 1-layer NN |
| | Epoch | 100 (50) | 500 (500) | 300 (300) | 200 (100) | 180 (180) |
| Common Parameters | $\gamma$ | 1 (1) | 5 (1) | 1 (1) | 1 (1) | 4 (4) |
| | $\alpha$ | 0.9 | 0.95 | 0.9 | 0.8 | - |
| | Batch size | 16 | 32 | 32 | 16 | 16 |
| | Optimizer | SGD | SGD | SGD | SGD | SGD |

## B.3 MORE EXPERIMENTS

### B.3.1 THE ACCURACY DISCREPANCY OF $f_{\text{DISK}}$ BETWEEN THE TRAINING AND VALIDATION SETS

Figure 8 visualizes the difference in predictive accuracy of $f_{\text{DISK}}$ between the training and validation sets. $f_{\text{DISK}}$ encourages a more pronounced spurious correlation to learn spurious information, resulting in an increase in the gap in predictive accuracy between the training and validation sets. This phenomenon is indeed observed in Figure 8, demonstrating the effectiveness of DISK.

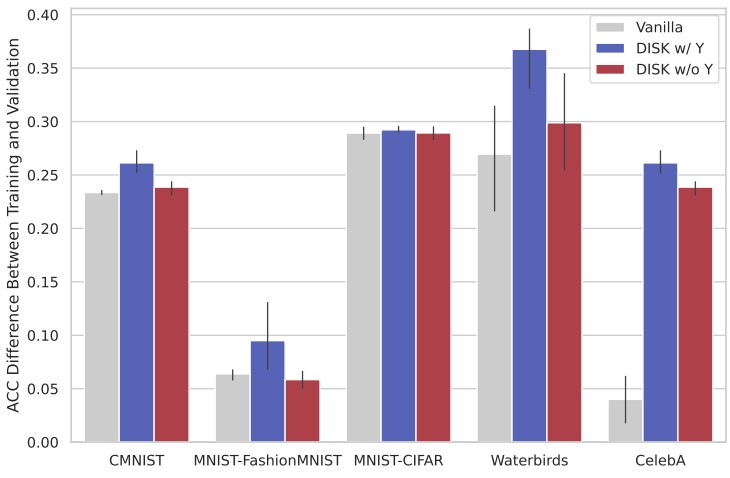

Figure 8: The difference in predictive accuracy of $f_{\text{DISK}}$ between the training and validation sets. We observe that DISK indeed encourages an increase in the gap in predictive accuracy between the training and test sets by learning spurious information, resulting in poorer generalization.

Table 5: Average Minority Domain Inference Precision $P_{\mathrm{M}}$ on four datasets without HCS. We observe that even without the HCS operation, DISK maintains high precision in recognizing challenging-to-identify minority groups. Although there is a slight decrease in accuracy compared to when HCS operation is applied, this suggests that DISK's performance in recognizing minority groups and inferring domain information primarily originates from its inherent capabilities.

| Method | CMNIST | MNIST FashionMNIST | MNIST CIFAR | WaterBirds |
|---|---|---|---|---|
| DISK $w/\ y^{val}$ | 81.2±1.9 | 80.4±4.7 | 86.9±1.1 | 35.8±4.0 |
| DISK $w/o\ y^{val}$ | 73.7±0.9 | 77.7±4.5 | 86.9±2.4 | 51.5±2.9 |

### B.3.2 ABLATION STUDY

Table 2 demonstrates that the HCS operation assists DISK in better identifying minority groups. To further understand the factors influencing DISK's domain partitioning ability, we present ablation experiment results in Tables 4 and 5, without the HCS operation, on the CMNIST, MNIST-FashionMNIST, MNIST-CIFAR, and Waterbirds datasets (the CelebA dataset did not undergo the HCS operation). We can observe that even without the HCS operation, DISK still significantly improves average/worst performance, regardless of the availability of $y^{val}$. This suggests that while HCS helps DISK better identify hard-to-recognize minority group samples, DISK's ability to partition domain information originates from its own core capabilities.

Table 4: Results of the ablation experiments. Even without HCS, significant improvements in performance relative to ERM are still observed for DISK, indicating that DISK's domain partitioning ability stems from its core capabilities instead of the HCS operation. The CelebA dataset is not included because CelebA does not use HCS.

| Method | HCS | CMNIST | | MNIST FashionMNIST | | MNIST CIFAR | | WaterBirds | |
|---|---|---|---|---|---|---|---|---|---|
| | | Avg. | Worst | Avg. | Worst | Avg. | Worst | Avg. | Worst |
| DISKS $w/\ y^{val}$ | ✓ | 65.1±1.7 | 67.6±2.0 | 92.3±0.8 | 92.6±0.9 | 69.0±0.4 | 69.2±0.6 | 91.1±1.4 | 85.5±3.0 |
| DISKS $w/o\ y^{val}$ | ✓ | 62.5±4.4 | 65.5±3.0 | 91.8±2.8 | 93.0±2.7 | 68.1±1.2 | 68.4±1.2 | 80.8±1.5 | 81.1±0.4 |
| DISKS $w/\ y^{val}$ | ✗ | 60.9±0.9 | 60.1±1.0 | 90.5±2.2 | 90.5±2.3 | 66.3±1.7 | 65.9±2.1 | 81.9±3.8 | 75.8±4.6 |
| DISKS $w/o\ y^{val}$ | ✗ | 57.7±5.6 | 56.7±6.5 | 85.7±2.9 | 85.5±3.0 | 66.2±2.5 | 65.7±3.0 | 80.8±2.1 | 70.1±2.5 |

### B.3.3 THE STABILITY OF DISK

The challenge addressed by Lin et al. (2022) revolves around testing the stability of domain inference algorithms. Specifically, it questions whether domain inference methods can learn invariant models from heterogeneous data originating from multiple environments with unknown environmental indices, aiming to demonstrate their stability. In the context of CMNIST, digits are considered as invariant features ($\mathbf{x}_1$), while colors are regarded as spurious features ($\mathbf{x}_2$). A variant of CMNIST, known as MCOLOR, has been created, where color assumes the role of the invariant feature, and digit shape serves as the spurious feature. The joint distribution $\mathbb{P}(\mathbf{x}_1, \mathbf{x}_2, y)$ of MCOLOR and CMNIST remains identical. The only distinction lies in the fact that CMNIST treats digits as the invariant feature, color as the spurious feature, and the prediction target is digit prediction. In contrast, MCOLOR treats color as the invariant feature, digits as the spurious feature, and the prediction target is color prediction. To underscore the poorer generalization of MCOLOR in this experiment, we retained only the digits 0 and 1 for both CMNIST and MCOLOR, eliminating the need for binary processing.

Table 6: Results on CMNIST and MCOLOR for DISK. DISK improves the performance of ERM on the test data for both CMNIST and MCOLOR, even when the data distributions are identical. This demonstrates that DISK can reliably identify spurious information for domain information inference.

| Method | CMNIST | | MCOLOR | |
|---|---|---|---|---|
| | Train Acc(%) | Test ACC(%) | Train Acc(%) | Test ACC(%) |
| ERM | 89.0±0.7 | 31.1±1.7 | 79.6±1.2 | 59.8±0.4 |
| DFR | 61.8±6.1 | 66.1±1.8 | 65.4±3.0 | 68.1±0.5 |
| DISKS $w/ y^{val}$ | 58.6±3.1 | 62.8±3.1 | 63.0±3.1 | 63.9±1.2 |
| DISKS $w/o y^{val}$ | 58.1±2.1 | 62.1±1.6 | 57.3±0.9 | 60.6±0.7 |

Table 6 also reports the training and test accuracy of DISK on CMNIST and MCOLOR. We observe that compared to CMNIST, MCOLOR exhibits fewer severe generalization issues, with an average test accuracy that approaches 60%. This suggests that color might be a feature easier to learn than digit shapes. We also notice that DISK significantly improves the performance of ERM on the test set for both CMNIST and MCOLOR, particularly when $y^{val}$ is available. This underscores the stability of DISK in domain information inference.

### B.3.4 DISK WITH MIXUP

To demonstrate that DISK can be combined with additional enhancement techniques, we present the performance of DISK and Mixup (DISKM) using the MNIST-FashionMNIST, and Waterbirds datasets as examples. Similar to LISA, Mixup includes two strategies: Intra-label Mixup (interpolating samples with the same label from different domains) and Intra-domain Mixup (interpolating samples from the same domain but different instances) (Yao et al., 2022). LISA is essentially a method that applies Mixup based on oracle domain information, representing the upper performance limit achievable by DISKM if the oracle domain information is accurate enough. Figure 9 describes the algorithmic process of DISKM for enhancing generalization. For Mixup, we sample the interpolation ratio parameter from a Beta$(2, 2)$ distribution, as recommended by LISA. Table 7 illustrates that DISKM significantly improves the accuracy of ERM, demonstrating the potential of combining DISK with various enhancement techniques.

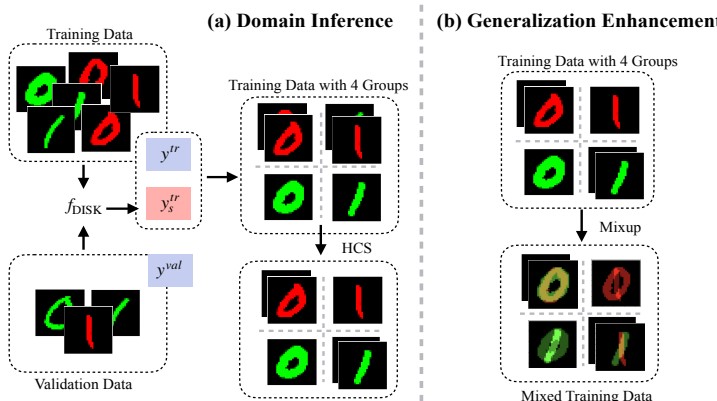

Figure 9: (a) DISK discovers the spurious correlation between the training and validation data, assigning spurious labels $y_s^{tr}$ to training instances. Subsequently, the training set is partitioned into different groups with distinct distributions $\mathbb{P}(\mathbf{x}_s, \mathbf{x}_v|g)$ where $g = (y, y_s)$. The HCS operation aids DISK in achieving a more precise inference of minority groups; (b) The training data from different domains undergo enhancement techniques, such as Mixup, which involves mixing up images across domains or across labels for further training.

Table 7: Comparison of Accuracy (%). DISKM achieves performance improvements compared to ERM, effectively addressing the challenge of weak ERM generalization without relying on domain information. We adopt the experimental results as reported in the original LISA paper by default, with the exception of the blue results for LISA on the Waterbirds dataset, which were obtained from our own experiments.

| Method | Domain Info | MNIST FashionMNIST | | WaterBirds | |
| | | Avg. | Worst | Avg. | Worst |
| --- | --- | --- | --- | --- | --- |
| ERM | $\times$ | $71.1\pm2.0$ | $69.6\pm2.0$ | $63.4\pm4.0$ | $34.4\pm8.2$ |
| IRM | $\checkmark$ | - | - | $87.5\pm0.7$ | $75.6\pm3.1$ |
| GroupDRO | $\checkmark$ | - | - | $91.8\pm0.3$ | $90.6\pm1.1$ |
| LISA | $\checkmark$ | $92.9\pm0.7$ | $92.6\pm0.8$ | $78.2\pm0.3$ / $91.8\pm0.3$ | $78.0\pm0.2$ / $89.2\pm0.6$ |
| DISKM $_{w/\ y^{val}}$ | $\times$ | $94.3\pm0.4$ | $94.1\pm0.5$ | $78.5\pm0.7$ | $77.8\pm0.1$ |
| DISKM $_{w/o\ y^{val}}$ | $\times$ | $93.6\pm1.2$ | $94.7\pm1.2$ | $78.5\pm0.3$ | $78.1\pm0.5$ |

### B.3.5 MORE VISUALIZATIONS ON WATERBIRDS

In this section, we present additional instances of Minority Group and Majority Group inferred by DISK, and investigate the similarity of DISK's partitioning when different random number seeds are used.

First, we visualize 40 instances of Minority Groups inferred by DISK under the same experimental settings, with the random seed set to 2, as in Section 4.2.2. In Figures 10, 11, 12 and 13 We observe that the reasons for Land (or Water) being misclassified as Water (Land), consistent with the findings in Section 4.2.2, are due to the fact that Land backgrounds, which are actually Land, contain typical Water features such as extensive blue areas (sky), which might lead DISK to categorize them as Water, which also has extensive blue regions like oceans. Conversely, Water backgrounds, which are actually Water, contain typical Land features such as vertical linear structures (tree branches or trunks or water ripples) and green tree leaves, which might lead DISK to classify them as Land. These findings, in conjunction with the results in Section 4.2.2, reveal that DISK infers domains based more on the similarity of underlying patterns rather than the patterns themselves.

Without loss of generality, we also observed the Minority Groups and Majority Groups inferred by DISK when the random seed is set to 0. We observed consistent results, as emphasized in Section 3.4, that DISK's instance recognition accuracy for Majority Groups is relatively higher than for Minority Groups. In the randomly selected 8 images, there were no misclassified samples within the Majority Group. However, for the misclassified samples within the Minority Groups, we found that it was still due to Land backgrounds containing extensive blue areas (sky) or water-like patterns, or Water backgrounds containing numerous land features, such as trees and tree branch reflections, which exhibit vertical stripe structures. By visualizing the Minority Groups and Majority Groups samples partitioned by DISK under different random seeds, we once again realize that DISK categorizes images based on the inherent similarity of their patterns. This neural network-based perspective of partitioning may introduce discrepancies compared to human decisions.

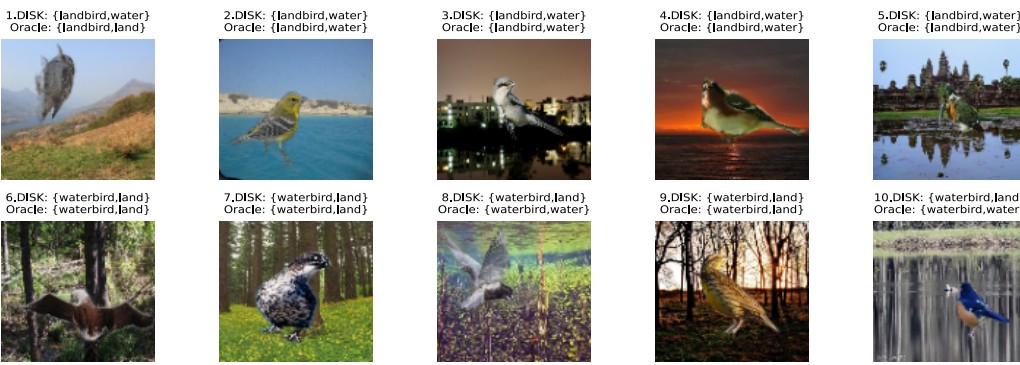

Figure 10: Comparison between DISK-inferred and oracle minority domains. We observe that the reason for Figure 1 being misclassified as Water by DISK is consistent with the misclassification reasons shown in Figure 5. In both cases, it is due to the background consisting of vast blue skies (a feature of water) and land. The misclassification of Figures 8 and 9 as Land instead of Water by DISK is also attributed to the presence of green foliage and extensive tree branches.

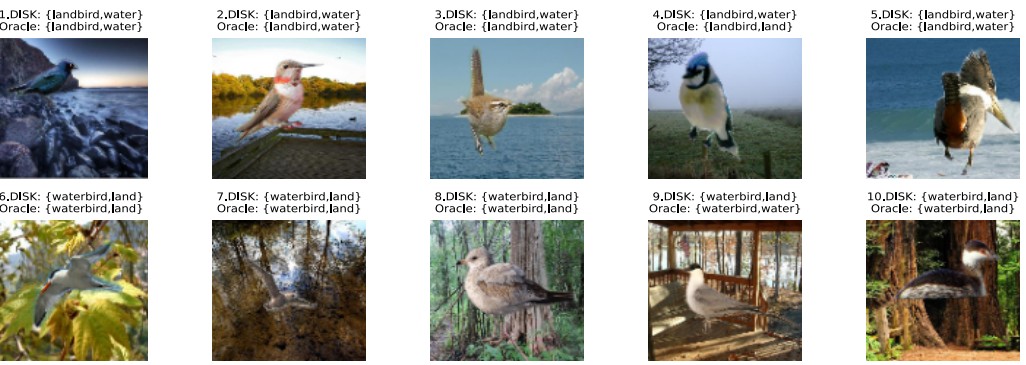

Figure 11: Comparison between DISK-inferred and oracle minority domains. We observe that the reason Figure 4 is misclassified by DISK as Water instead of Land is due to its background consisting of extensive blue skies (a characteristic of water) and land. Similarly, Figure 9 is misclassified by DISK as Land instead of Water because it contains a significant number of tree branches (linear structures).

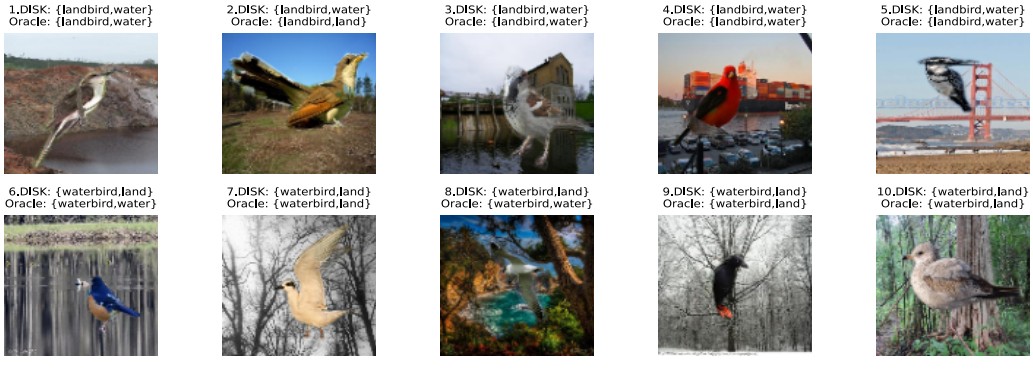

Figure 12: Comparison between DISK-inferred and oracle minority domains. We observed that the reason for Figure 2 being misclassified by DISK as Water instead of Land is due to its background consisting of extensive blue sky (a characteristic of water). Similarly, Figures 6 and 8 being misclassified as Land by DISK instead of Water is attributed to the presence of numerous tree branches (linear structures) and green trees, which are typical land features.

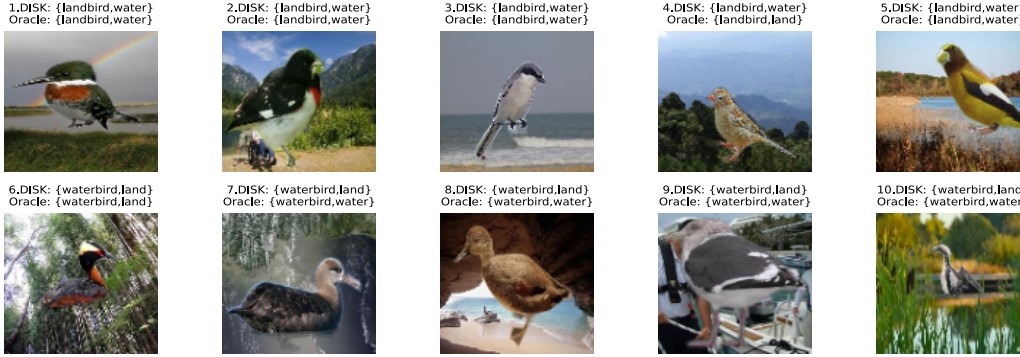

Figure 13: Comparison between DISK-inferred and oracle minority domains. We observe that the reason Figure 4 is misclassified by DISK as Water instead of Land is due to its background featuring extensive blue sky (a characteristic of water). Figures 7, 8, 9, and 10 being misclassified as Land instead of Water by DISK are attributed to the presence of numerous linear structures resembling tree branches in the images, or the absence of prominent water features (unlike Figures 1, 2, 3, 5, where backgrounds typically include extensive blue areas; even in the case of Figures 8 and 9, the backgrounds lack extensive blue areas and are dominated by yellow land, people, and clutter often seen on land).

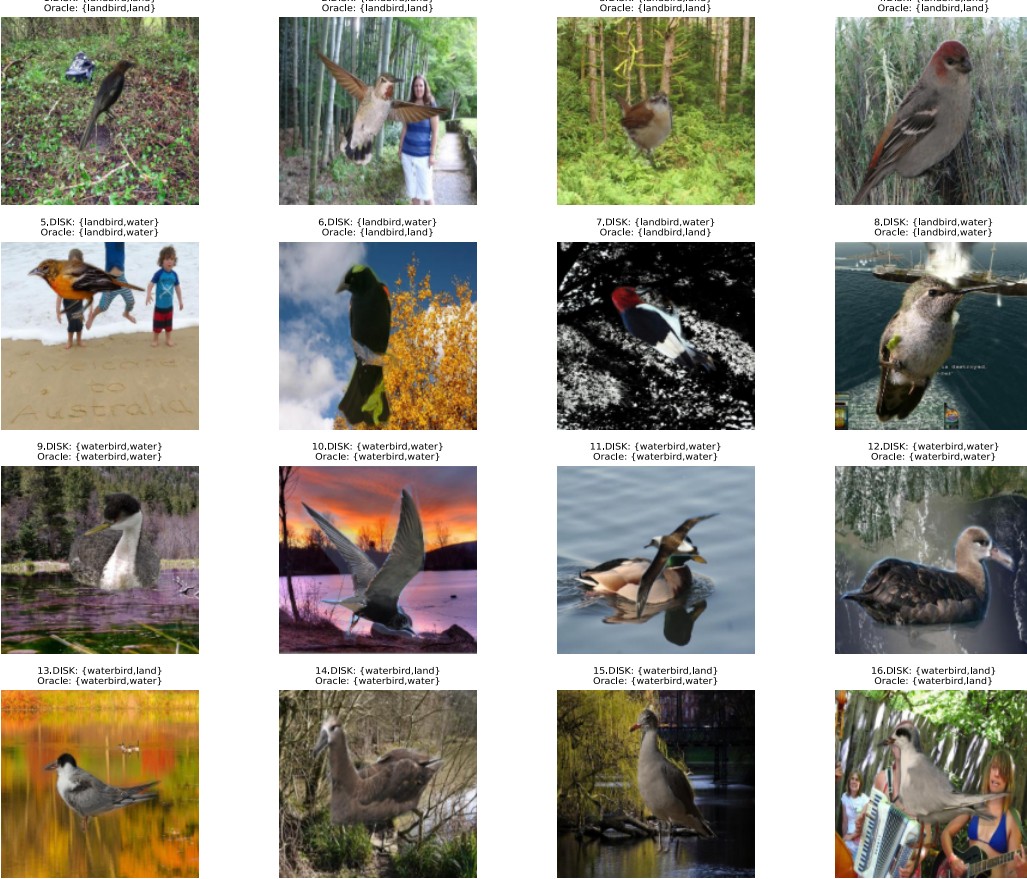

Figure 14: Comparison between DISK-inferred and oracle minority domains. We observe that Figures 6 and 7 are misclassified as Water by DISK due to their backgrounds consisting of extensive blue skies (a water-related feature) or horizontally striped branches that resemble rippling water surfaces, which can be confusing for DISK. On the other hand, Figures 13, 14, and 15 are misclassified as Land by DISK because they contain a significant amount of vertical linear structures, tree elements, tree reflections, and other land-related features.

