# OpenReview forum: "DISK: Domain Inference for Discovering Spurious Correlation with KL-Divergence"
_ICLR.cc/2024/Conference — Submitted to ICLR 2024_

### Official Review · Reviewer_eWm8 · 2023-11-01

**Soundness:** 2 fair
**Presentation:** 2 fair
**Contribution:** 2 fair
**Rating:** 3
**Confidence:** 4

**Summary:**

This paper presents an approach, DISK, for estimating the spurious label of a data sample to enable the use of downstream algorithms that can then use the estimated group information to address subpopulation shifts in the training data. The methodology is developed to leverage an important premise that the imbalance among subpopulations (in training data) is absent in the validation data, based on which a spurious classifier is optimized to maximize the correlation of the estimated spurious label and task label while minimizing such correlation.  To demonstrate the effective of DISK, the estimated spurious label is then used to enable a subsampling strategy to balance subgroups, with a strange of HCS to discard low-confidence predictions of spurious labels.

Experiments were conducted on a synthetic toy case and benchmark MNIST, F-MNIST, and CIFAR datasets. Comparisons were made primarily to baselines that require actual spurious labels. Specifically, in comparison with DFR, the authors demonstrated the performance of DISK to be similar without needing the use of the true spurious label.

**Strengths:**

- To remove the need of actual spurious labels is an important challenge in addressing subpopulation shifts.
- The in-depth analysis of the ability of DISK to infer spurious labels and its consistency with actual oral decision is appreciated.

**Weaknesses:**

Three of my major concerns are listed below:

1. The motivation of DISK is that it will remove the need of the knowledge of the data labels regarding their subgrouping based on core and spurious features. At the same time, the methodology is built on the premise that there is a training set where such spurious correlation exits, along with a validation set where such spurious correlation does not exist. It seems to me that this premise itself requires knowledge about the subgroups during data curation (in order to create validation data where subgroups are balanced) — this seems to be contradictory to the said motivation.

2. The experiments were mostly focused on comparison with baselines that requires the label of subgroups — because DISK does not use such labels, its performance is expected to not exceed these baselines, as summarized in Table 1. While these experiments are useful to show the DISK was able to produce reasonable results without the use of spurious labels, there are however a large number of works that had proposed alternative ways of estimating the spurious labels, such as clustering based approaches ([1][2], loss based approaches ([3][4][5]  — these should be the primary baselines to demonstrate the contribution of DiSK.

[1] No Subclass Left Behind: Fine-Grained Robustness in Coarse-Grained Classification Problems
[2] Unsupervised Learning of Debiased Representations with Pseudo-Attributes
[3] Just Train Twice: Improving Group Robustness without Training Group Information
[4] Learning from Failure: Training Debiased Classifier from Biased Classifier
[5] SPREAD SPURIOUS ATTRIBUTE: IMPROVING WORST-GROUP ACCURACY WITH SPURIOUS ATTRIBUTE ESTIMATION

3. Throughout the introduction and the related work sections, the description of the problem of spurious correlation due to subpopulation distribution shifts is heavily mixed with domain adaptation and generalization problems, as if they were equivalent. This misses the important distinction between these two sets of problem despite their connection: in the former, the domain label and core label are heavily correlated due to the subpopulation distribution, versus in the general DA/DG setting, there is typically no such correlation. The authors can be more rigorous and careful in these discussion to make that distinction so as to not confuse the audience.

**Questions:**

I’d like to see my major comments regarding 1-2 be addressed.

In addition,
1. It is not clear how cross-entropy H and how the conditional density in equation (6) or (4) are calculated.
2. In (4)-(6), y^tr/^val refers to the label of core features, and y_s,w^tr/^val refers to the label of spurious features, is this true? This was not clearly defined.
3. I’d imagine that sub-sampling would see a major limitation when facing substantially underrepresented groups. The use of HCS, by further discarding low-confidence predictions, will only make it worse. Please comment on this, in particular the limit of subgroup imbalance ratio DISK can handle properly.
4. It’d be very helpful in the toy data experiments to see the estimated spurious labels.

---

> ### Author Response · Authors · 2023-11-16
> **Response to Reviewer eWm8**
>
> 【**Q1**】 The motivation of DISK is that it will remove the need of the knowledge of the data labels regarding their subgrouping based on core and spurious features. At the same time, the methodology is built on the premise that there is a training set where such spurious correlation exits, along with a validation set where such spurious correlation does not exist. It seems to me that this premise itself requires knowledge about the subgroups during data curation (in order to create validation data where subgroups are balanced) — this seems to be contradictory to the said motivation.
>
> 【**A1**】There may be some misunderstanding here.
>
> 1. Actually, we did not assume that the spurious correlation does not exist in the validation dataset. Our algorithm is designed to effectively mitigate the spurious correlations as long as these correlations appear slightly different between the training and validation sets. If the spurious correlations remain consistent across both sets, our algorithm would reduce to the standard ERM approach.
>
>    We then present the following two pieces of evidence to refute the point that a "validation set where such spurious correlation does not exist."
>
> - Firstly, taking our experimental data as examples, as elaborated in Appendix B.2.1 DATASET DETAILS, it is evident that only the validation sets of CMNIST and Waterbirds data do not exhibit spurious correlations (i.e., the distribution of spurious features within the same class is balanced). However, for MNIST-CIFAR, MNIST-FashionMNIST, and CelebA, the validation sets all demonstrate varying spurious correlations (similar to the training set, the distribution of spurious features within the same class remains unbalanced). Nevertheless, DISK has shown improved performance across all datasets.
>
> - Secondly, we present additional experimental results in the table below to illustrate whether DISK is effective or not does not depend on the existence or severity of spurious correlations in the validation set. As long as there the spurious correlations between the training set and the validation set appear (slightly) difference, DISK can still be effective.
>
>   We specifically manipulate the distribution of spurious features (Woman : Man) within the blonde hair group (class 1) in the validation set to gradually deviate from the training set, where the ratio of spurious features for class 1 is 94.3:5.7.
>
>   We observed that although the validation set in our paper (the ratio is 94.0:6.0) which exhibits very close spurious correlations to those in the training set (the ratio is 94.3:5.7), DISK results in an increase in ERM from 38.0 to 64.8.  When we slightly increase the disparity between the validation set and the training set (e.g., adjusting the ratio to 90:10), DISK's performance further improves. This once again demonstrates the effectiveness of DISK, which is completely independent of whether there are spurious correlations in the validation set or whether it is balanced.
>
>   - **Additional experimental results on CelebA** (seed = 0, the ratio is 94.3:5.7 in training set）
>
>   | Ratio of  Spurious feature (Woman : man) for class 1 | Average ACC    | Worst ACC      |
>   | ---------------------------------------------------- | -------------- | -------------- |
>   | 94.0:6.0 (Table 1 in our paper)                          | 88.8 $\pm$ 0.3 | 64.8 $\pm$ 1.3 |
>   | 90.0:10.0                                              | 87.4           | 68.8           |
>   | 80.0:20.0                                             | 88.1           | 73.5           |
>   | 70.0:30.0                                                | 87.9           | 78.3           |
>   | 60.0:40.0                                                | 88.6           | 75.0           |
>   | 50.0:50.0                                                | 86.1           | 73.1           |
>
> 2. Since we do not require the validation set to be balanced, concerning the construction of the validation set, as we have previously mentioned in the model section (Page 5), it is apparent that a natural (unlabeled) validation set can be directly constructed using the data queried during the test phase, without any additional requirements. As long as the spurious correlations in the test phase differ from those in the training phase, the proposed algorithm can successfully identify the spurious correlation and enhance generalization. This ability of DISK is completely unrelated to whether the subgroups in the validation set are balanced or not. This construction method does not necessitate any knowledge of the spurious correlation and can be easily performed.
>
> We hope that the above clarifications can alleviate the confusion you may have had. If you have any further questions, we are more than happy to answer them.

---

> ### Author Response · Authors · 2023-11-16
> **Response to Reviewer eWm8**
>
> 【**Q2**】The experiments were mostly focused on comparison with baselines that requires the label of subgroups — because DISK does not use such labels, its performance is expected to not exceed these baselines, as summarized in Table 1. While these experiments are useful to show the DISK was able to produce reasonable results without the use of spurious labels, there are however a large number of works that had proposed alternative ways of estimating the spurious labels, such as clustering based approaches ([1][2], loss based approaches ([3][4][5] — these should be the primary baselines to demonstrate the contribution of DiSK.
>
> 【**A2**】
>
> 1. For baselines like  LfF, we have already discussed in the main text (page 3 RELATED WORK section and page 7 Baselines section) why they are not considered as baselines. We want to reiterate that previous research (Lin, Yong, et al.NeurIPS 2022) has shown that  LfF rely solely on the training dataset and require invariant information, which leads to instability in detecting spurious features. For example, Tabel 1 in the research (Lin, Yong, et al.NeurIPS 2022) reported that  LfF completely fail to handle the challenges of CMNIST and MCOLOR datasets, achieving only 21.2±0.4 test accuracy for LfF on CMNIST . We also applied DISK in the CMNIST and MCOLOR datasets in the EXPERIMENTS section (page 6) and in section B.3.3 THE STABILITY OF DISK. We compared DISK with LfF to observe if DISK would face the same issues as LfF. Table 6 in section B.3.3 THE STABILITY OF DISK demonstrates that, relative to LfF, DISK does not fail on CMNIST and MCOLOR and still remains good performance when invariant information is unavailable.
>
>
> 2. For baselines like JTT,  they may require a stronger assumption as it utilizes the group label (that is not needed in our approach) from the validation set to select parameters and determine the effectiveness of identifying spurious correlation during the identification stage   (Liu, Evan Z., et al. ICLR2021;Zhang, Michael, et al. ICML2022 ;Kirichenko, P., et al ICLR2023).  So to ensure fairness, we compared DISK with JTT, which does not use any group label information from the validation set, on the WaterBirds dataset. We observe that when no group label information from the validation set was used, DISK outperformed JTT on worst group accuracy. We will include these results in the revised version.
>
>   - **Comparison on Waterbirds**. Both DISK and JTT do not use any validation group label information (repeat each method three times with seed = 0,1,2).
>
>   | Methods            | Average ACC    | Worst ACC      |
>   | ------------------ | -------------- | -------------- |
>   | JTT                | 97.7 $\pm$ 0.1 | 69.3 $\pm$ 3.0 |
>   | DISK w $y^{val}$   | 91.1 $\pm$ 1.4 | 85.5 $\pm$ 3.0 |
>   | DISK w/o $y^{val}$ | 80.8 $\pm$ 1.5 | 81.1 $\pm$ 0.4 |
>
> 3. Finally, we would like to highlight that, as mentioned in our paper (page 7, Baselines section), DFR which uses oracle group labels  represents the upper limit of effectiveness of the DISK with subsampling strategy. The results presented in Table 1 indicate that DISK is already in close proximity to DFR and even slightly outperforms DFR on waterbirds. This strongly demonstrates the effectiveness of DISK. And for the remaining baselines, we will also provide additional results in the revised version.
>
> 【**Q3**】The authors can be more rigorous and careful in these discussion to make that distinction so as to not confuse the audience.
>
> 【**A3**】In the introduction section, we clarify the common definition of subpopulation shifts and domain (Yao, Huaxiu, et al. ICML2022) by words and examples (page 1, figure 1). Furthermore, in the method section (section 3.1 PRELIMINARIES, section 3.2 FORMAL DEFINITION OF SPURIOUS LABELS), we once again emphasize the connection between the concepts of spurious/invariant, domain, and subpopulation (Yang, Yuzhe, et al., 2023) using our mathematical notion. If these concepts still confuse you, we will further emphasize and clarify them in the new version. Thank you for your suggestion.

---

> ### Author Response · Authors · 2023-11-16
> **Response to Reviewer eWm8**
>
> 【**Q4**】 It is not clear how cross-entropy H and how the conditional density in equation (6) or (4) are calculated.
>
> 【**A4**】As demonstrated in the method section (page 5) and Appendix 1, the cross-entropy H is indeed the cross-entropy between the spurious label $\hat y_s$ and true label $y$.
>
> For the conditional density in equation (6) or (4), we utilize the property of KL divergence to transform the conditional density into the joint density. Taking Equation (4) as an example (the same treatment applies to Equation (6)):
>
> $$
> KL(P( y^{tr} | \hat y^{tr}_{s,w}) || P(y^{val} |  \hat y^{val}\_{s,w} )) =  KL(P( y^{tr} ,\hat y^{tr}\_{s,w} )|| P( y^{val}, \hat y^{val}\_{s,w} )) - KL(P(\hat y^{tr}\_{s,w} )|| P(\hat y^{val}\_{s,w} ))
> $$
>
> We then estimate the KL divergence using the MINE algorithm (Belghazi, Mohamed Ishmael, et al.  ICML2018). For more details, please refer to the provided code.
>
> 【**Q5**】 In (4)-(6), $y^{tr}$, $y^{val}$ refers to the label of core features, and $y_{s,w}^{tr}$ $y_{s,w}^{val}$ refers to the label of spurious features, is this true? This was not clearly defined.
>
> 【**A5**】Invariant features are attributes (such as digit contours in the color-MNIST dataset) that consistently have a causal relationship with the true label $y^{tr}$ or $y^{val}$ across all data (training, validation, and test sets). So, $y^{tr}$ and $y^{val}$ refer to the labels of the core features.
>
> From a causal perspective, spurious attributes (such as colors in the color-MNIST dataset) are defined as attributes that are not causally related to the true label but are correlated with the true label $y^{tr}$ in the training data due to data sampling bias or imbalance. Based on this concept, we introduce spurious labels to represent the labels assigned by the classifier based on the instance's spurious features.
>
> 【**Q6**】I’d imagine that sub-sampling would see a major limitation when facing substantially underrepresented groups. The use of HCS, by further discarding low-confidence predictions, will only make it worse. Please comment on this, in particular the limit of subgroup imbalance ratio DISK can handle properly.
>
> 【**A6**】Firstly, it is important to note that prioritizing high-confidence samples does not necessarily lead to a situation where things  "only get worse ".  As discussed in Section 3.4, the model's predictions for minority samples are not reliable, and it frequently misclassifies majority samples as minority samples with low confidence. The purpose of high confidence selection (HCS) is to eliminate these majority samples while retaining the minority samples. Our ablation experiments in Section 4.2.2 and Appendix B.3.2 further validate our claim that HCS effectively enhances the performance of the model.
>
> And for "the limit of subgroup imbalance ratio DISK can handle properly", as mentioned in our response in 【**A2**】, actually, three out of the five datasets used in our experiments had validation sets with totally imbalanced subgroup . However, DISK still demonstrated excellent performance across all five datasets. Furthermore, we conducted additional experiments on CelebA to illustrate that as long as there are (slight) differences between the spurious relationships in the validation set and what the training set exhibits, DISK can effectively handle it. This effectiveness is completely unrelated to whether the subgroups in the validation set are balanced or not. Please refer to the response in【**A2**】for more details.
>
> 【**Q7**】It’d be very helpful in the toy data experiments to see the estimated spurious labels.
>
> 【**A7**】We provide the accuracy results of the DISK classifier in predicting spurious labels on toy data as follows. The results demonstrate that DISK achieves a high accuracy in predicting spurious labels, confirming its ability to classify instances based on the spurious features.
>
> | Method              | Train Accuracy | Eval Accuracy | Test Accuracy |
> | ------------------- | -------------- | ------------- | ------------- |
> | DISKS w $y^{val}$   | 98.7           | 96.0          | 96.6          |
> | DISKS w/o $y^{val}$ | 95.0           | 95.5          | 96.4          |

---

### Official Review · Reviewer_cYfB · 2023-11-03

**Soundness:** 3 good
**Presentation:** 2 fair
**Contribution:** 2 fair
**Rating:** 6
**Confidence:** 4

**Summary:**

To address subpopulation shift without domain information, the authors propose DISK, a novel and effective end-to-end method for domain inference. The advantages are demonstrated on some datasets.

**Strengths:**

1. This work is relatively solid. Details of necessary proofs and experiments are attached
2. The authors focus on domain inference, providing insights on domain inference, illustrating that when spurious features contain complex information. The comparison of the neural network-based DISK and human decisions on potential to capture the essence of the data is also discussed.
3. Experiments are relatively well-designed, including the experiments on synthetic 2D data and real-world data.

**Weaknesses:**

1. The available and details could be discussed in more detail. The label or domain is the core of this work. When the data is labeled with this information, some carry it, and none carry it at all, is there a difference in the way the model works
2. About the formal definitions of spurious labels, technical details are ignored. Specifically, how to obtain and why consider both the spurious representation and the invariant representation. What is the advantage of such processing over other simpler method, like clusters and others?
3. Differences in data distribution may affect the effectiveness of the model. Will the discussion of discrepancy between the correlations in the training and validation set in spurious term leads to data leakage? Furthermore, data distribution seems important to the effectiveness of the model, then how to ensure the universality and wide effectiveness of DISK?
4. The representation of this paper could be optimized. For example, Related work is better to be better organized. Then naturally, the motivation and advantages of this work over previous ones could be more obvious. Furthermore, figures are lacked, especially for the ones describing the framework.
5. Some modules are lack of theorical basis. For example, why High Confidence Selection (HCS) is effective. And how to define the high-confidence and corresponding threshold when mitigating subpopulation shift with DISK.
6. The symbols used throughout this paper should be gathered, presented in the form of tables.
7. Experiments are not sufficient enough. Typically, ablation study on the strategy of determining spurious labels, the necessity of utilizing KL-Divergency and so on.
8. Experiments do not fully demonstrate the advantages of DISK. For example, the advantages of it over baselines are not evident. Furthermore, the analysis on its performance on datasets like Celeb are not convincing. It is more like a kind of guess, and if so, DISK seems lack of universality.

**Questions:**

1. When the data is labeled with this information, some carry it, and none carry it at all, is there a difference in the way the model works
2. How to obtain and why consider both the spurious representation and the invariant representation. What is the advantage of such processing over other simpler method, like clusters and others?
3. Will the discussion of discrepancy between the correlations in the training and validation set in spurious term leads to data leakage? Furthermore, data distribution seems important to the effectiveness of the model, then how to ensure the universality and wide effectiveness of DISK?
4. Could you summarize the motivation and contributions of this work? And could the workflow of DISK be summarized in the form of figure?
5. Why High Confidence Selection (HCS) is effective. And how to define the high-confidence and corresponding threshold when mitigating subpopulation shift with DISK. Corresponding analysis could be more theorical and solid.
6. Why the advantages of DISK over baselines are not evident. Furthermore, the analysis on its performance on datasets like Celeb are not convincing. It is more like a kind of guess, and if so, DISK seems lack of universality. Could you demonstrate the university of DISK?

---

> ### Author Response · Authors · 2023-11-16
> **Response to Reviewer cYfB**
>
> 【**Q1**】The available and details could be discussed in more detail...is there a difference in the way the model works
>
> 【**A1**】 Thanks for your question.  We require a labeled training set and a natural (unlabeled) validation set, which can be constructed directly using the data obtained during the testing phase.
>
> From a causality perspective, spurious attributes (e.g., colors of the color-MNIST dataset)  are defined as the attributes that are not causally related to the truth label , but are correlated with the truth label  in the training data due to data sampling bias or imbalance. And, invariant features refer to attributes (e.g., digit contours of the color-MNIST dataset) that are consistently causally related to the true label across all (training/validation/test) data. Based these concepts,  we introduce the  spurious labels to represent the labels assigned by the classifier based on the instance's spurious features, while the true features are used for classification based on the instance's invariant features.
>
> Models trained on the training data can successfully classify based on spurious features instead of invariant features , thus result in a high correlation between the spurious labels and the true labels on the training data. However, the degree of correlation between the spurious labels and the true labels in the validation set may differ. DISK mitigates the influence of spurious features by measuring this difference between the spurious correlation of training set and the validation set.
>
> 【**Q2**】 About the formal definitions of spurious labels... What is the advantage of such processing over other simpler method, like clusters and others?
>
> 【**A2**】  Please refer to the definition of spurious labels provided in 【**A1**】.
>
> Since spurious and invariant representations are coupled in the data representation(obtained from a feature extractor, e.g., ResNet on the data x), there is no need to explicitly obtain them. Instead, we can use DISK to train a classifier $f_{disk}$ on the data representation and obtain spurious labels based solely on the spurious representation.
>
> For why we did not consider clustering methods,
>
> (1) Firstly, accurate clustering is a challenging task for high-dimensional data and must deal with the curse of dimensionality (Steinbach, et al. bioinformatics).
>
> (2)  Secondly,  both spurious features and invariant features in the data are inherently coupled. Simple clustering alone cannot guarantee that instances are clustered based on spurious features, thus making it impractical to utilize clustering results for further mitigation of the spurious correlation.
>
> 【**Q3**】Differences in data distribution may affect the effectiveness of the model... then how to ensure the universality and wide effectiveness of DISK?
>
> 【**A3**】  Thanks for your question.
>
> We evaluate the performance of the model on the test set, not the training set or validation set. Therefore, there is no issue of data leakage.
>
> And for the impact of the data distribution of the validation set on DISK,
>
> (1) Firstly, take our experimental data as examples, for MNIST-CIFAR, MNIST-FashionMNIST and CelebA,  the distribution differences between the training and validation sets are not very significant and DISK still can improve the prediction accuracy on test compared to ERM. In particular, for the CelebA dataset, both the validation set and the training set demonstrate a very similar distribution. However, DISK was still able to improve the prediction accuracy of the worst group from 38.0 to 64.8.
>
> (2)  Secondly, we provide additional experimental results in the table below to demonstrate DISK's sensitivity to the difference in the validation set. We only adjust the distribution of spurious features within the blonde hair group (class 1) in the validation set to make it gradually different from the training set (the ratio of spurious features (Woman : Man) for class 1 in the training data is 94.3:5.7).
>
> We observe that when we slightly increase the difference between the validation set and the training set (e.g., 90:10), DISK's performance further improves.
>
> - **Additional experimental results on CelebA** （seed = 0, the ratio is 94.3:5.7 in training set）
>
> | Ratio of  Spurious feature (Woman : man) for class 1 | Average ACC    | Worst ACC      |
> | ---------------------------------------------------- | -------------- | -------------- |
> | 94.0:6.0 (Table 1 in our paper)                      | 88.8 $\pm$ 0.3 | 64.8 $\pm$ 1.3 |
> | 90.0:10.0                                            | 87.4           | 68.8           |
> | 80.0:20.0                                            | 88.1           | 73.5           |
> | 70.0:30.0                                            | 87.9           | 78.3           |
> | 60.0:40.0                                            | 88.6           | 75.0           |
> | 50.0:50.0                                            | 86.1           | 73.1           |

---

> ### Author Response · Authors · 2023-11-16
> **Response to Reviewer cYfB**
>
> 【**Q4 and Q6**】 The representation of this paper could be optimized. For example, Related work is better to be better organized. Then naturally, the motivation and advantages of this work over previous ones could be more obvious. Furthermore, figures are lacked, especially for the ones describing the framework. The symbols used throughout this paper should be gathered, presented in the form of tables.
>
> 【**A4 and A6**】 Thank you for your suggestion. We will make these revisions to highlight the advantages of DISK and add a table of symbols in the revised version.
>
> 【**Q5**】Some modules are lack of theoretical basis. For example, why High Confidence Selection (HCS) is effective. And how to define the high-confidence and corresponding threshold when mitigating subpopulation shift with DISK.
>
> 【**A5**】Regarding the effectiveness of High Confidence Selection (HCS), as discussed in Section 3.4, the model tends to make inaccurate predictions for minority samples, and predictions classified as minority samples often include majority samples with low confidence. HCS is designed to address this issue by selectively removing majority samples while preserving minority samples. Our ablation experiments (Section 4.2.2 and Appendix B.3.2) further validate our claim that HCS contributes to improving model performance.
>
> As for the selection of the threshold $\alpha$, as shown in Table 3, a rough selection of $\alpha$ around 0.9 can yield desirable results.
>
> 【**Q7**】Experiments are not sufficient enough. Typically, ablation study on the strategy of determining spurious labels, the necessity of utilizing KL-Divergency and so on.
>
> 【**A7**】KL divergence is a common measure for quantifying differences between distributions and has been extensively employed in many researches. In our study, we specifically opted for KL divergence due to the following reasons:
>
> (1) The distributions being compared share the same support, making KL divergence capable of providing reasonable measures.
>
> (2) Existing research  (Belghazi, Mohamed Ishmael, et al.  ICML2018) indicates that estimating KL divergence using the MINE algorithm is a straightforward and easy process.
>
> Although exploring different distribution metrics and their potential impact on performance is an interesting topic, it goes beyond the scope of our paper, which primarily focuses on introducing a method for identifying spurious correlations. The investigation of alternative metrics is left for future research.
>
> 【**Q8**】Experiments do not fully demonstrate the advantages of DISK. For example, the advantages of it over baselines are not evident. Furthermore, the analysis on its performance on datasets like Celeb are not convincing. It is more like a kind of guess, and if so, DISK seems lack of universality.
>
> 【**A8**】  Perhaps there is some misunderstanding. As shown in Table 1, we observed that for all experimental data, DISK without using any group label information exhibits a significant improvement over ERM, while for the baselines (IRM, GroupDRO, LISA, DFR) that use group label information, DISK shows performance that is quite close. These findings are sufficient to demonstrate the significant improvement of DISK over the baselines.   Furthermore, we want to stress that, as previously stated in the paper (page 7 Baselines section), DFR utilizing oracle group labels represents the maximum effectiveness of DISK with a subsampling strategy. As indicated by the findings in Table 1, DISK is already approaching DFR and even slightly surpasses DFR in the case of waterbirds. This serves as compelling evidence of the effectiveness of DISK.
>
> For CelebA data, we provide additional results in response【**A3**】to support our view: DISK has significantly improved performance  compared to ERM.  The experimental results in 【**A3**】also indicate that if we could, like some existing methods such as JTT (Liu, Evan Z., et al. ICLR2021), utilize some additional group label information from the validation set and slightly adjust the distribution of the validation set, the improvement of DISK would be more pronounced.

---

### Official Review · Reviewer_eBHc · 2023-11-05

**Soundness:** 3 good
**Presentation:** 3 good
**Contribution:** 2 fair
**Rating:** 3
**Confidence:** 4

**Summary:**

The authors study the problem of tackling spurious correlations during supervised learning, in the setting where spurious labels are not known during training. They propose a method to infer the spurious label given a validation set where spurious correlations differ, which can then be used in a second stage training to learn an unbiased model (e.g. with subsampling). Their method involves learning some spurious label that maximizes mutual information with the class label on the training set, while having the maximum divergence when conditioned on with the validation set. They evaluate their method on a variety of datasets, finding that they outperform the baselines.

**Strengths:**

- The paper tackles an important real-world problem.
- The proposed method is intuitive.
- The paper is well-written and easy to understand.

**Weaknesses:**

1. The authors should formally state some of the assumptions required for the method to learn the correct spurious label. It seems like the assumption informally is that the true spurious label is the feature that is strongly correlated with the label while giving the maximum divergence on the validation set. If this is the case, how would the method behave in the presence of multiple spurious correlations [1]?

2. The authors should empirically and theoretically characterize how different the validation dataset has to be from the training dataset in order for the method to effectively learn the spurious label.

3. The proposed method assumes access to an unlabelled validation set with different spurious correlations from the training, which is not a very strict assumption. However, the method also implicitly assumes prior knowledge of the number of possible values that the spurious correlation can take, in order to learn $w$. Prior methods like JTT [2] do not have this assumption.

4. The authors should compare against more baselines which also do not require knowledge of the spurious label during training, such as JTT [2] or CnC [3]. They should also compare both performance and identified minority groups against prior group discovery method such as EIIL.

5. The authors should evaluate their method on additional benchmarking datasets in the NLP domain such as MultiNLI and CivilComments.

6. The proposed metric (Definition 2) seems a bit flawed, as the model could infer very few minority samples to get a high value. The authors should consider using a metric that balances precision and recall such as the intersection-over-union, and also reporting the inference accuracy.

[1] A Whac-A-Mole Dilemma: Shortcuts Come in Multiples Where Mitigating One Amplifies Others. CVPR 2023.

[2] Just Train Twice: Improving Group Robustness without Training Group Information. ICML 2021.

[3] Correct-n-Contrast: A Contrastive Approach for Improving Robustness to Spurious Correlations. ICML 2022.

**Questions:**

Please address the weaknesses and answer the following questions:
1. What was the strategy for hyperparameter selection (i.e. to find the hyperparameters in Table 3)? Presumably, this requires access to a labelled validation set?

2. What was the validation set used in each of the datasets? How different were they from the training set?

3. How sensitive is the method to the choice of $\gamma$? The authors should consider showing the WGA when sweeping over this hyperparameter.

4. Have the authors tried additional methods for the 2nd stage other than subsampling, such as GroupDRO? Does this improve performance further?

5. In Figure 3, second row, it might be clearer to additionally report the accuracy of the classifier in distinguishing the spurious correlation, as that is what matters the most in this row.

---

> ### Author Response · Authors · 2023-11-16
> **Response to Reviewer eBHc**
>
> 【**Q1**】The authors should formally state some of the assumptions ... It seems like the assumption informally is that the true spurious label is the feature that is strongly correlated with the label while giving the maximum divergence on the validation set. If this is the case, how would the method behave in the presence of multiple spurious correlations?
>
> 【**A1**】In fact, we can show that our algorithm can effective mitigate the spurious correlation when the spurious correlations appear (slightly) different in training and validation set. Thus such a difference between the spurious correlations in training and validation sets can be implicitly viewed as the requirement to predict the correct spurious label.  Please refer to our response in 【**A2**】, which provides more information to demonstrate that as long as there are (slight) differences in the spurious correlations between the validation set and the training set, DISK can be effective.
>
> Furthermore, our method can indeed handle multiple spurious correlation. In this case, the spurious feature will correspond to the combinations of these spurious correlations, i.e., all spurious correlations will contribute to the DISK framework to help predict the spurious label.
>
> 【**Q2**】The authors should empirically and theoretically characterize how different the validation dataset has to be from the training dataset in order for the method to effectively learn the spurious label.
>
> 【**A2**】For the difference between the training data and validation dataset,
>
> (1) Firstly, take our experimental data as examples, as shown in Appendix B.2.1 DATASET DETAILS, for MNIST-CIFAR, MNIST-FashionMNIST and CelebA,  the distribution differences between the training and validation sets are not very significant and DISK still can  improve the prediction accuracy on test compared to ERM. In particular, for the CelebA dataset, both the validation set and the training set exhibit (very close) spurious correlations in class 1. However, DISK still managed to improve the prediction accuracy of the worst group from 38.0 to 64.8.
>
> (2)  Secondly, we provide additional experimental results in the table below to demonstrate DISK's sensitivity to the difference between the training data and the validation data. We only change the distribution of spurious features (Woman : Man)  within the blonde hair group (class 1) in the validation set to make it gradually different from the training set t(the distribution of spurious features for class 1 in the training data is 94.3:5.7).
>
> We observed that although the validation set in our paper which exhibits very close spurious correlations to those in the training set, DISK results in an increase in ERM from 38.0 to 64.8. Furthermore, when we slightly increase the difference between the validation set and the training set (e.g., 90:10), DISK's performance improves.
>
> - **Additional experimental results on CelebA** （seed = 0, the ratio is 94.3:5.7 in training set）
>
> | Ratio of  Spurious feature (Woman : man) for class 1 | Average ACC    | Worst ACC      |
> | ---------------------------------------------------- | -------------- | -------------- |
> | 94.0:6.0 (Table 1 in our paper)                      | 88.8 $\pm$ 0.3 | 64.8 $\pm$ 1.3 |
> | 90.0:10.0                                            | 87.4           | 68.8           |
> | 80.0:20.0                                            | 88.1           | 73.5           |
> | 70.0:30.0                                            | 87.9           | 78.3           |
> | 60.0:40.0                                            | 88.6           | 75.0           |
> | 50.0:50.0                                            | 86.1           | 73.1           |
>
> 【**Q3**】The proposed method assumes access to an unlabelled validation set with different spurious correlations from the training...However, the method also implicitly assumes prior knowledge of the number of possible values that the spurious correlation can take. Prior methods like JTT do not have this assumption.
>
> 【**A3**】We don't need to know the number of possible values that the spurious correlation can take. In fact, we set the number of spurious labels to be exactly the same as the number of data labels, then the values that the spurious correlation will be categorized into these spurious labels via the DISK training (i.e., one spurious label can take multiple spurious correlations).
>
> Furthermore, for methods like JTT, it may require a stronger assumption as it utilizes the group label (that is not needed in our approach) from the validation set to select parameters and determine the effectiveness of identifying spurious correlation during the identification stage (Liu, Evan Z., et al. ICLR2021;Zhang, Michael, et al. ICML2022 ;Kirichenko, P., et al ICLR2023). According to our experiments, if JTT is not provided with such a group label information from the validation set, its performance is worse than DISK (see the detailed results in response【**A4**】).

---

> ### Author Response · Authors · 2023-11-16
> **Response to Reviewer eBHc**
>
> 【**Q4**】The authors should compare against more baselines which also do not require knowledge of the spurious label during training, such as JTT or CnC. They should also compare both performance and identified minority groups against prior group discovery method such as EIIL.
>
> 【**A4**】
>
> 1. Regarding baselines like EIIL, we have already addressed their exclusion as baselines in the main text, specifically in the RELATED WORK section on page 3 and the Baselines section on page 7. It is important to reiterate that previous research (Lin, Yong, et al.NeurIPS 2022) has demonstrated the reliance of EIIL solely on the training dataset and the requirement of invariant information, which ultimately leads to instability in detecting spurious features. As an example, Table 1 in the aforementioned research (Lin, Yong, et al.NeurIPS 2022) reported that EIIL completely fails to handle the challenges presented by the CMNIST and MCOLOR datasets, achieving a test accuracy of only 17.8±0.4 on MCOLOR. In our own experiments, detailed in the EXPERIMENTS section on page 6 and section B.3.3 THE STABILITY OF DISK, we applied DISK to the CMNIST and MCOLOR datasets and also compared it with EIIL to observe if DISK encounters similar issues. Table 6 in section B.3.3 THE STABILITY OF DISK demonstrates that, in comparison to EIIL, DISK does not fail on CMNIST and MCOLOR and continues to exhibit good performance even when invariant information is unavailable.
>
> 2. When considering baselines like JTT and CnC, it is worth noting that they may require a stronger assumption due to their utilization of group labels from the validation set. These group labels are not necessary in our approach. Their usage involves selecting parameters and determining the effectiveness of identifying spurious correlations during the identification stage (Liu, Evan Z., et al. ICLR2021;Zhang, Michael, et al. ICML2022 ;Kirichenko, P., et al ICLR2023). To ensure fairness, we compared DISK with JTT, which does not utilize any group label information from the validation set, specifically on the WaterBirds dataset. We observed that when no group label information from the validation set was used, DISK outperformed JTT in terms of worst group accuracy. These results will be included in the revised version.
>
> - **Comparison on Waterbirds**. Both DISK and JTT do not use any validation group label information (repeat each method three times with seed = 0,1,2)
>
> | Methods            | Average ACC    | Worst ACC      |
> | ------------------ | -------------- | -------------- |
> | JTT                | 97.7 $\pm$ 0.1 | 69.3 $\pm$ 3.0 |
> | DISK w $y^{val}$   | 91.1 $\pm$ 1.4 | 85.5 $\pm$ 3.0 |
> | DISK w/o $y^{val}$ | 80.8 $\pm$ 1.5 | 81.1 $\pm$ 0.4 |
>
> 3. Finally, we would like to emphasize that, as we mentioned in the paper (page 7 Baselines section), DFR using oracle group labels is the upper limit of the effectiveness of DISK with subsampling strategy. According to the results shown in Table 1, DISK is already very close to DFR and even slightly outperforms DFR on waterbirds. This strongly demonstrates the effectiveness of DISK.
>
> 【**Q5**】The authors should evaluate their method on additional benchmarking datasets in the NLP domain such as MultiNLI and CivilComments.
>
> 【**A5**】Thank you for your suggestion. We will include more experimental results in the revision.
>
> 【**Q6**】The proposed metric (Definition 2) seems a bit flawed, as the model could infer very few minority samples to get a high value. The authors should consider using a metric that balances precision and recall such as the intersection-over-union, and also reporting the inference accuracy.
>
> 【**A6**】 We propose the Minority Domain Inference Precision based on the following reasons:
>
> Firstly, focusing on high-confidence samples does not necessarily lead to "the model could infer very few minority samples to get a high value." As mentioned in Section 3.4, the model's predictions for minority samples are inaccurate, and samples predicted as minority samples often include majority samples with low confidence. High confidence selection (HCS) aims to remove these majority samples without reducing minority samples. Our ablation experiments (Section 4.2.2 and Appendix B.3.2) also support our viewpoint that HCS indeed helps improve model performance.
>
> Secondly, similar to the typical definition of precision, the precision we propose describes the model's effectiveness in identifying minority samples. We found that the improvement in precision, relative to the recall metric, is more important for creating a  balanced new training set for ERM retraining.
>
> Therefore, we believe that the proposed metric can better help identify the true minority samples and thus lead to performance gain, especially when the dataset is extremely unbalanced.

---

> > ### Author Response · Authors · 2023-11-16
> > **Response to Reviewer eBHc**
> >
> > 【**Q7 and Q9**】What was the strategy for hyperparameter selection (i.e. to find the hyperparameters in Table 3)? Presumably, this requires access to a labelled validation set? How sensitive is the method to the choice of $\gamma$? The authors should consider showing the WGA when sweeping over this hyperparameter.
> >
> > 【**A7 and A9**】 As stated in our experimental setup (page 8, Model Training), we strictly followed the experimental settings of (Yao, Huaxiu, et al. ICML2022) to determine the neural network architectures and hyperparameters.
> >
> > For $\gamma$, we used a grid search approach to determine its value. We set a search space of [1, 2, 3, 4, 5] and selected the best-performing $\gamma$. The choice of $\gamma$ is not sensitive since a small search space from 1 to 5 can yield excellent performance across all experimental data and we observed that for most datasets in our experiments, $\gamma$ can be fixed at 1 directly (Table 3).
> >
> > It should be emphasized that all parameter selections in our experiments do not require access to a validation set with labels and group labels.
> >
> > 【**Q8**】What was the validation set used in each of the datasets? How different were they from the training set?
> >
> > 【**A8**】 Regarding the construction of validation set, we have mentioned this in the model section (Page 5). It can be seen that a natural (unlabeled) validation set can be constructed directly using the data queried in the test phase.
> >
> > And please see the【**A2**】for the question about difference.
> >
> > 【**Q10**】Have the authors tried additional methods for the 2nd stage other than subsampling, such as GroupDRO? Does this improve performance further?
> >
> > 【**A10**】 We have already mentioned that in addition to the subsampling strategy, we also considered the combination of DISK and Mixup  (Page 7, 4.2 REAL-WORLD DATA). The experimental results in the appendix demonstrate that even the combination of DISK with simple strategies like subsampling and Mixup can yield significant improvements. In the revised versions, we will include the results of DISK with GroupDRO.
> >
> > 【**Q11**】In Figure 3, second row, it might be clearer to additionally report the accuracy of the classifier in distinguishing the spurious correlation, as that is what matters the most in this row.
> >
> > 【**A11**】  We report the accuracy of the DISK classifier in distinguishing the spurious correlation, as shown in the table below. DISK exhibits high accuracy in predicting spurious labels, indicating that DISK indeed classifies instances based on the spurious features.
> >
> > |                               | Train Accuracy | Eval Accuracy | Test Accuracy |
> > | ----------------------------- | -------------- | ------------- | ------------- |
> > | DISK with $y^{val}$    | 98.7           | 96.0          | 96.6          |
> > | DISK without  $y^{val}$ | 95.0           | 95.5          | 96.4          |

---

### Official Review · Reviewer_R9Uq · 2023-11-06

**Soundness:** 2 fair
**Presentation:** 2 fair
**Contribution:** 2 fair
**Rating:** 3
**Confidence:** 3

**Summary:**

This paper presents an approach called DISK for domain inference. Specifically, the DISK approach infers domains by finding a dataset partition that maximizes the predictive difference between the training set and a validation set. The spurious correlation differs between the training and validation sets.

**Strengths:**

This paper tried to tackle an important and extremely challenging problem: domain inference without any domain information.

**Weaknesses:**

1. The assumption of a validation dataset does not seem weaker than an assumption on auxiliary information. This paper assumes that the spurious correlation in the training set does not hold in the validation dataset. Without explicitly knowing the spurious correlation, the construction of such a validation set is non-trivial.

2. The motivation behind the KL maximization objective (Equation 2) is unclear. This objective aims to maximize the label distribution difference between inferred domains, but the spurious correlation does not necessarily relate to the label distribution. For example, one domain may have 50% red zero digits and 50% blue one digits, and the other domain has 50% blue zero digits and 50% red one digits. Here, the label distribution is the same for the two domains.

3. The definition of spurious label is confusing. Definition 1 states that a spurious label indicates the category of an instance. However, the abstract says that the spurious label is also the domain label. The domain label does not indicate the category of an instance.

4. The steps of the proposed approach are not described. How do we get the spurious label $y_s$ from $f_{DISK}$?

5. The experiment section seems incomplete. Is IRM inapplicable to the MNIST and CIFAR experiments?

**Questions:**

1. Is there any particular reason to use KL divergence in your algorithm? There are many other candidates such as Wasserstein distance.

2. What is the performance of baselines under the DISK setting? Here, we can use the training set and the validation set as two domains. No exact domain information is needed.

---

> ### Author Response · Authors · 2023-11-16
> **Response to Reviewer R9Uq**
>
> **【Q1】** This paper assumes that the spurious correlation in the training set does not hold in the validation dataset. Without explicitly knowing the spurious correlation, the construction of such a validation set is non-trivial.
>
> **【A1】**  Sorry for making the confusion. In fact, we did not assume that the spurious correlation does not hold in the validation dataset. Our algorithm can effective mitigate the spurious correlation as long as the spurious correlations appear differently in training and validation set, otherwise it will reduce to standard ERM.  Actually, based on our experimental data, specifically detailed in Appendix B.2.1 DATASET DETAILS, it is clear that only the validation sets of CMNIST and Waterbirds datasets do not display spurious correlations. However, for MNIST-CIFAR, MNIST-FashionMNIST, and CelebA, the validation sets exhibit varying degrees of spurious correlations. DISK has consistently demonstrated improved performance across all datasets. In particular, for the CelebA dataset, both the validation set and the training set exhibit severe spurious correlations in class 1. However, DISK still managed to improve the prediction accuracy of the worst group from 38.0 to 64.8.
>
> Regarding the construction of validation set, there are no specific requirements for constructing the validation set since there is no need to know any spurious correlations within it. As we have mentioned this in the Method section,  it can be seen that a natural (unlabeled) validation set can be constructed directly using the data queried in the test phase. Then as long as the spurious correlations in the test phase is different from that in the training phase, the proposed algorithm can successfully identify the spurious correlation and then improve the generalization. This construction does not need the knowledge of spurious correlation, and thus can be trivially performed.
>
> 【**Q2**】The motivation behind the KL maximization objective (Equation 2) is unclear. This objective aims to maximize the label distribution difference between inferred domains, but the spurious correlation does not necessarily relate to the label distribution.
>
> 【**A2**】There might be a misunderstanding, the difference between the conditional distributions (conditioned on the spurious label) in Equation 2 can be large. Let's still focus on your example, where the spurious label can be red or blue. Then given the spurious label to be red, the label distribution in the first domain will be {0: 100%; 1: 0%}; while in the second domain, the label distribution will be {0: 0%; 1: 100%}. Then it is clear that these two label distributions are very different.
>
> **【Q3】** The definition of spurious label is confusing.
>
> **【A3】** Sorry for the confusion, it is true that the spurious label is different from the domain label. In fact, spurious label refers to the label assigned to an instance by a classifier based on spurious features (e.g., colors of the color-MNIST dataset), which appear strongly in the data distribution but exhibit a spurious correlation with the true labels.  We will provide further clarification on this in the new version.
>
> **【Q4】** The steps of the proposed approach are not described. How do we get the spurious label $y_s$ from $f_{disk}$?
>
> **【A4】** In fact, by training using the DISK objective (Equation 6), we can get the parameter $w$, which is exactly the weights of the $f_{disk}$, i.e., the model of the spurious classifier. Then applying $f_{disk}$ to the representation of the data, i.e., $z$ (obtained from a feature extractor on $x$, e.g., ResNet), we can get the spurious label $y_s$ for the data $x$. We will summarize the proposed approach explicitly in the revision.
>
> **【Q5】** The experiment section seems incomplete. Is IRM inapplicable to the MNIST and CIFAR experiments?
>
> **【A5】** The results of IRM on MNIST-FashionMNIST and MNIST-CIFAR are as follows. We observe that  IRM effectively mitigates the performance degradation on the test set caused by spurious correlations relative to ERM. We will add this experimental result in the new version.
>
> - IRM on MNIST-FashionMNIST and  MNIST-CIFAR. Seed =0 ,1,2
>
> | Dataset            | Average ACC  | Worst ACC    |
> | ------------------ | ------------ | ------------ |
> | MNIST-FashionMNIST | 88.1$\pm$0.8 | 81.9$\pm$0.7 |
> | MNIST-CIFAR        | 78.4$\pm$0.9 | 73.2$\pm$1.1 |

---

> > ### Author Response · Authors · 2023-11-16
> > **Response to Reviewer R9Uq**
> >
> > 【**Q6**】 Is there any particular reason to use KL divergence in your algorithm? There are many other candidates such as Wasserstein distance.
> >
> > 【**A6**】 KL divergence is a common measure used to quantify the difference between distributions, and has been widely used in many papers. Regarding Wasserstein distance, which is typically used when the supports of two distributions are different. However, in our case, the distributions being compared have the same support so that KL divergence can already provide reasonable measures. It is indee interesting to try different distribution metrics and they might lead to different performance, but this is beyond the scope of this paper (our paper is to provide a method to identify the suprious correlation) and we leave it to our future work.
> >
> > 【**Q7**】What is the performance of baselines under the DISK setting? Here, we can use the training set and the validation set as two domains. No exact domain information is needed.
> >
> > 【**A7**】  We have already included baselines in our experimental results, but we suspect that this might be different from the *baselines under the DISK setting* you mentioned. Can you be more specific, like to provide more description about the DISK setting. We are happy to answer in detail if you can provide more descriptions.

---

> ### Comment · Reviewer_R9Uq · 2023-11-20
> **Reply to Authors**
>
> Thanks for replying.
>
> [A1] In the response, "the spurious correlation as long as the spurious correlations appear differently in training and validation set" implies the spurious correlation in the training set does not hold on the validation set.
>
> [A4] Adding an algorithm that describes the steps of DISK may help.
>
> [A6] This is hand-waving. An ablation study may help.
>
> [A7] The IRM algorithm can access two domains (i.e., training and validation sets in the paper) where the spurious correlations appear differently.

---

> ### Author Response · Authors · 2023-11-21
> **Response to Reviewer R9Uq**
>
> Thank you for your prompt reply. For your questions,
>
> **[A1]**  There may be still have some misunderstanding.
> - Firstly, our algorithm is capable of effectively mitigating spurious correlations as long as they manifest in slight differences in the training and validation sets. Otherwise, it will reduce to standard ERM, which will not harm downstream tasks. The effectiveness of this "slightly different" phenomenon is clearly demonstrated in CelebA. In the case of CelebA, we observed that even though the validation set (with a spurious features ratio of 94.0:6.0) exhibited very similar spurious correlations to those in the training set (with a spurious features ratio of 94.3:5.7), the use of DISK resulted in an increase in ERM from 38.0 to 64.8.
>
> - Secondly, we would like to emphasize that it is **not a strict assumption** to require slight differences in the spurious correlations between the training and validation sets (The Reviewer eBHc shares the same opinion as us that *The proposed method assumes access to an unlabelled validation set with different spurious correlations from the training, which is not a very strict assumption.*).
> And **this is evident in many existing methods that utilize validation sets with different spurious correlations, and some even directly rely on group labels in the validation set to improve model performance** (Yoonho Lee et al., ICLR2023; Liu, Evan Z. et al., ICLR2021; Nam, J. et al., ICLR2021, Michael Zhang et al., ICML2021). Our method also aims to achieve good results with different (even slightly different) validation sets, without the need for group label information, and this does not constitute a strict assumption.
>
> **[A4]**  Thank you for your suggestion. We will add more explanations as requested. In the original main text on page 4, we directly present the formulation $\hat y_{s,w} = f_{disk}(z^{tr},w)$ in order to show how to obtain $\hat y_{s,w}$ using $f_{disk}$. We hope this can provide you with a clear understanding of the process.
>
> **[A6]**  We would like to reiterate that the purpose of this paper is to present a method for detecting spurious correlations. We consider KL-divergence because it satisfies the support of the distributions being compared and it is natural to use KL or cross-entropy as the loss when the inputs are soft labels and labels.  And previous studies have used KL-divergence (Yoonho Lee et al., ICLR2023) to measure distribution divergence and alleviate spurious correlations. Using Wasserstein distance may also yield valid results, but it deviates from the standard approach. Additionally, it is not possible to list all possible measures of distribution divergence, and finding the best measure is beyond the scope of this paper.
>
> However, we will explore the use of Wasserstein distance in our algorithm to investigate whether it can outperform KL divergence or not. If you could provide relevant references regarding the benefits of Wasserstein distance over KL divergence, we would greatly appreciate it. This would inspire and motivate us to further investigate and understand the differences between the two in our task.
>
> **[A7]** There may be some misunderstanding. IRM and our method are designed to address completely different problems with different settings.
>
> - IRM assumes knowledge of the environmental information, where the correlation between spurious features and labels may vary across different environments, while the correlation between invariant features and labels remains constant. IRM aims to learn a feature representation that solely depends on the invariant features. However, the acquisition of environmental information is a challenging and highly important problem, and our method (including methods discussed in our related work) aims to address this issue.
> - Specifically, we do not have access to environmental information or knowledge of spurious features. Instead, we aim to learn the spurious features by detecting spurious correlation, allowing us to assign samples to different groups/environments. The information obtained from this grouping can be further combined with subsequent methods, such as subsampling, mixup, or even IRM.

---

### Meta-Review · Area_Chair_qZNw · 2023-12-05

**Metareview:**

This paper proposes an information theoretical objective for domain inference. The method DISK in eq.(3) maximizes the mutual information between the true label and the spurious label in the training set, and at the same time maximizes the discrepancy between spurious correlation in the training and validation set.
On MNIST/CIFAR datasets, it shows more accurate domain inference and better generalization.

Strength:

This paper tackles an important real-world problem and the proposed objective makes sense.

Weaknesses:

- Although the paper is motivated by intuitive examples, the basic settings, the set of assumptions, and the definitions are not clearly stated. This makes the contributions hard to parse. Many questions about the basic assumptions/definitions are raised by the reviewers, hinting that they can be better explained in the text.

- Experimentally, as suggested by the reviewers, this work can benefit from another round of revision to include more baselines, sensitivity study, and more fair experimental design. The authors already did some additional experiments in the rebuttal but they should be sufficiently verified and reorganized.

The reviewers provided many constructive comments. The authors mentioned that more details of the proposed approach, and more experimental results will be included in the revision. However, the paper is not revised accordingly in the rebuttal period (which is understandable due to limited time). These changes will require more time and effort on the authors' side.

**Justification For Why Not Higher Score:**

Based on the reviews, this paper needs major rewriting to improve the technical quality and the experimental design.

**Justification For Why Not Lower Score:**

N/A

---

### Decision · Program_Chairs · 2024-01-16

Reject